# Network mechanisms and dysfunction within an integrated computational model of progression through mitosis in the human cell cycle

**Scott S. Terhune**[1,2]☯*, **Yongwoon Jung**[2]☯, **Katie M. Cataldo**[1], **Ranjan K. Dash**[2,3]*

**1** Departments of Microbiology and Immunology, Medical College of Wisconsin, Milwaukee, Wisconsin, United States of America, **2** Department of Biomedical Engineering, Medical College of Wisconsin, Milwaukee, Wisconsin, United States of America, **3** Department of Physiology, Medical College of Wisconsin, Milwaukee, Wisconsin, United States of America

☯ These authors contributed equally to this work.
* sterhune@mcw.edu (SST); rdash@mcw.edu (RKD)

**Data Availability Statement:** All relevant data are within the manuscript and its Supporting Information files.

## Abstract

The cellular protein-protein interaction network that governs cellular proliferation (cell cycle) is highly complex. Here, we have developed a novel computational model of human mitotic cell cycle, integrating diverse cellular mechanisms, for the purpose of generating new hypotheses and predicting new experiments designed to help understand complex diseases. The pathogenic state investigated is infection by a human herpesvirus. The model starts at mitotic entry initiated by the activities of Cyclin-dependent kinase 1 (CDK1) and Polo-like kinase 1 (PLK1), transitions through Anaphase-promoting complex (APC/C) bound to Cell division cycle protein 20 (CDC20), and ends upon mitotic exit mediated by APC/C bound to CDC20 homolog 1 (CDH1). It includes syntheses and multiple mechanisms of degradations of the mitotic proteins. Prior to this work, no such comprehensive model of the human mitotic cell cycle existed. The new model is based on a hybrid framework combining Michaelis-Menten and mass action kinetics for the mitotic interacting reactions. It simulates temporal changes in 12 different mitotic proteins and associated protein complexes in multiple states using 15 interacting reactions and 26 ordinary differential equations. We have defined model parameter values using both quantitative and qualitative data and using parameter values from relevant published models, and we have tested the model to reproduce the cardinal features of human mitosis determined experimentally by numerous laboratories. Like cancer, viruses create dysfunction to support infection. By simulating infection of the human herpesvirus, cytomegalovirus, we hypothesize that virus-mediated disruption of APC/C is necessary to establish a unique mitotic collapse with sustained CDK1 activity, consistent with known mechanisms of virus egress. With the rapid discovery of cellular protein-protein interaction networks and regulatory mechanisms, we anticipate that this model will be highly valuable in helping us to understand the network dynamics and identify potential points of therapeutic interventions.

**Funding:** This work was supported by the Advancing Healthier Wisconsin, Research and Education Program grant 5520429 to SST and RKD, and by National Institutes of Allergy and Infectious Disease division of the National Institutes of Health under award number R01AI083281 to SST. ST and RD were funded by NIH R21 grant R21-AI149039-01. The content is solely the responsibility of the authors and does not necessarily represent the official views Advancing Healthier Wisconsin or the National Institutes of Health. The funders had no role in study design, data collection and analysis, decision to publish, or preparation of the manuscript.

**Competing interests:** The authors have declared that no competing interests exist.

## Author summary

The knowledge of cellular protein-protein interaction networks and their regulatory mechanisms governing cell cycle is ever expanding at an extraordinary rate. This includes many publications defining small and disconnected subsets of functional interactions occurring at limited time points. For cellular mitosis, mechanisms underlying the process have been experimentally investigated for several decades. However, we do not have an integrated quantitative understanding of mitosis and relative contributions of different regulatory mechanisms under normal conditions and their dysregulations in diseases. Our goal is to develop an in silico simulation of human mitosis using published experimental data by integrating subsets of mechanistic relationships into a single base computational model with enough resolution to approximate outcomes upon perturbations. In achieving this goal, we have developed a novel comprehensive computational model that simulates the human mitotic cell cycle and provides an integrated quantitative understanding of how human mitosis is altered during disease. We have suitably defined model parameter values and tested the model to reproduce the cardinal features of human mitosis determined experimentally by numerous laboratories. The developed model will be highly valuable in helping us to understand complex network relationships, build new hypotheses, design new experiments, and identify points of therapeutic interventions.

## Introduction

Disruption of cell cycle is an element of almost all diseases including cancer and viral infections. In the context of viral infections, viral proteins manipulate the host cell cycle constructing pseudo-cycles to support virion particle replication (reviewed in [1]). Cell cycle consists of oscillating changes in protein concentrations and activities within the cell allowing for duplication of the host genome and eventual cell division. This process is highly regulated involving multiple cellular mechanisms directly participating in the process and others that sense the cellular environment. These include oscillating changes in protein expressions regulated at the transcriptional and post-transcriptional levels, changes in post-translational modifications and functional activities, and changes in protein-protein interactions and subcellular localizations (reviewed in [2–4]). Alteration in any one of these events can result in changes in the dynamics of each relationship, and ultimately in the cell cycle dysregulation and pseudo-cycle creation, a hallmark of cell pathobiology in cancer and viral infections.

The final stage in the cell cycle is mitosis resulting in cytokinesis and cell separation. Disruptions to regulation can result in an arrest and potentially mitotic collapse (i.e. loss in all oscillations) or catastrophe resulting in cell death [5]. The unidirectional process is centered upon activation and then inactivation of Maturation Promoting Factor (MPF), which is a complex in mammalian cells of Cyclin B1 (CCNB1) binding to Cyclin-dependent kinase 1 (CDK1) [2]. MPF kinase (CCNB1:CDK1) formation is initiated in the G2 phase by increasing steady state levels of CCNB1, dynamic changes in phosphorylation status, and relocalization of numerous proteins. These changes regulate a concentration-dependent switch-like mechanism. One critical residue on CDK1 in the MPF complex is tyrosine 15 which is phosphorylated by several kinases to CDK1P (P, phosphorylated) [6]. Modification by Wee1-like protein kinase (WEE1) disrupts MPF kinase activity and, in this study, we refer to this inactive protein complex as preMPF (CCNB1:CDK1P). Dephosphorylation of preMPF is mediated by Cell division cycle protein 25 phosphatases (CDC25). Activities of both WEE1 and CDC25

enzymes are also tightly regulated. Their regulators include Polo-like kinase 1 (PLK1) and several phosphatases (PPase) such as CDC14A and Protein phosphatase 2A (PP2A) [7–10]. Positive feedback by active MPF kinase contributes to further WEE1 inactivation and CDC25 activation. Upon reaching a threshold level, active MPF initiates nuclear envelope breakdown (NEBD), spindle formation, and chromatin condensation by phosphorylating diverse targets.

Transition through mitosis involves activation of the E3 ubiquitin-protein ligase complex known as Anaphase-promoting complex or cyclosome (APC/C) [3, 4, 11, 12]. The approximate nineteen-subunit ubiquitin ligase complex is stimulated by phosphorylation by MPF and PLK1 kinases and binding of activator subunits, Cell division cycle protein 20 (CDC20) and CDC20 homolog 1 (CDH1). Activator subunits are also regulated by phosphorylation and associations with regulatory proteins. For example, CDC20 is inhibited by MPF- and PLK1-mediated phosphorylation and by the Mitotic Checkpoint Complex (MCC), which senses microtubule-kinetochore assembly and tension. Following rapid dephosphorylation of CDC20 by PP2A phosphatase and establishment of high spindle tension, the APC/CP:CDC20 ubiquitin ligase complex becomes active targeting substrates for degradation. These include degradation of Securin (PTTG1), which prevents sister chromatid separation by the enzyme Separase (ESPL1), and degradation of CCNB1. The second activator, CDH1 is also inhibited by MPF- and PLK1-mediated phosphorylation while activated by a delayed PP2A-mediated dephosphorylation event. This process is temporally driven by PP2A and differential kinetics of substrate dephosphorylation between phosphoserine and phosphothreonine residues [13]. In addition to all of these relationships, crosstalk between several layers of regulation (e.g. DNA damage response, chromosome condensation, and kinetochore formation) involving diverse post-translational modifications (e.g. SUMOylation, acetylation, and methylation) contributes to the progression through mitosis [14]. Upon activation of the APC/C:CDH1 ubiquitin ligase complex, mitotic exit is initiated by degradation of substrates including CDC20, PLK1, and any remaining CCNB1.

Because of the enormous complexity and timing of relationships in the cell cycle, it is not always possible to identify key cellular mechanisms responsible for dysregulation relying on experiments alone. Integrated computational modeling has been instrumental in simulating and predicting outcomes upon perturbation of single or multiple cellular mechanisms. Several mathematical models have been constructed describing subsets of interactions occurring during different phases of the cell cycle including mitosis (e.g. see [15–34]). These studies have provided novel mechanistic insights into complex behaviors of multiple regulators (e.g. activators and inhibitors) during the cell cycle progression and mitosis. The initial models on MPF kinase activation and feedback regulations by WEE1 and CDC25 [15–17, 21, 24, 25] determined that the relationships act as a bistable on/off switch, committing cells to mitotic entry. CCNB1 levels and its localization contribute to the sensitivity of the regulated switch. A second bistable on/off switch regulates PP2A phosphatase [10, 27, 30, 32, 35, 36], which involves MPF kinase-mediated activation of human Greatwall kinase (hGWL). hGWL kinase phosphorylates substrates which, in turn, inhibit PP2A. Therefore, PP2A activity is inhibited during mitotic entry. The eventual dephosphorylation of hGWL allows for PP2A regulation of WEE1 kinase, CDC25 phosphatase, and APC/C. Computational modeling in combination with *in vitro* experimental studies have also revealed two CCNB1 concentration thresholds that drive mitosis [17, 21, 24, 25, 30, 36, 37]. The first high CCNB1 concentration threshold regulates ESPL1 (Separase) activation, and the second intermediate CCNB1 concentration allows for PP2A activation. Cumulatively, the bistable on/off switches involving MPF kinase and PP2A phosphatase are interconnected creating a mechanism that drives unidirectional and irreversible progression through mitosis.

Current computational models of the cell cycle are limited to only subsets of relationships (i.e. interacting reactions and their regulations) and are defined in a variety of experimental cell systems such as yeast and frog oocytes. For the purpose of generating new hypotheses and predicting new experiments designed to study human disease, we have developed here a novel integrated computational model for human cell cycle regulation focusing exclusively on the mitotic phase. Unique features include CCNB1 and PLK1 initiation of mitotic entry, transitions through APC/C:CDC20 and APC/C:CDH1 to mitotic exit, rate constants for synthesis, and rate constants for multiple mechanisms of degradation. Using a combination of previously published kinetic parameter values and analysis of qualitative and quantitative experimental data, we have parameterized and tested the model to reproduce the cardinal behaviors of several mitotic proteins occurring in human cell lines [7–9, 13, 38–40]. We have also used this newly constructed model to provide insights into the dysregulation and pseudo-cycle creation required for successful infection of a human herpesvirus, cytomegalovirus (HCMV). By integrating mechanisms defined in the literature, this unique model will be immensely valuable in helping the scientific community understand complex network relationships and build hypotheses relating to mitotic cell cycle dysregulation occurring in diverse pathologies.

## Results

### Model-simulated dynamic responses from an integrated interaction network of human mitotic regulators

Progression through mitosis is driven by a complex set of protein-protein interaction reactions and their regulations that change over time. These relationships have been experimentally defined over several decades using a variety of conditions, and each published study compares subsets of relationships at limited time points. Our goal was to develop an in silico simulation of human mitosis using published experimental data from human cells by integrating subsets of mechanistic relationships using both quantitative and qualitative data into a single base computational model with enough resolution to approximate outcomes upon perturbation. Mitosis, as described within the Introduction, begins upon MPF kinase (CCNB1:CDK1) activation and ends upon APC/C:CDH1-stimulated degradation of mitotic regulators (reviewed in [2–4]). We have defined the full cycle of the mitotic biopathway (Fig 1A) through biochemical reactions among mitotic proteins and associated protein complexes. These reactions describe associations and dissociations of proteins, which are influenced by concentrations and the rates of different reactions. Concentrations are further determined by the rates of protein synthesis and degradations. These are unique features of this novel model that are rarely addressed in other publications. In addition, we have included regulatory changes in phosphorylation and dephosphorylation. A complete description of each mitotic protein is provided in S1 Appendix, and that of biochemical reaction is provided in S2 Appendix. During cell cycle, relationships and reactions are continuously changing over time. We have captured the dynamics of mitotic proteins based on ordinary differential equations (ODEs). These ODEs are based on the principle of mass conservation for the proteins/protein complexes and include a hybrid Michaelis-Menten and mass action kinetic formulation for the biochemical reactions. The ODEs and model parameters are described in S3 Appendix and S4 Appendix, respectively. When combined, the resulting simulation of the mitotic biopathway (Fig 1A) includes 12 major mitotic proteins and associated protein complexes in 27 different states interacting by 15 major biochemical reactions governed by 26 ODEs. Simplified wiring diagrams show both positive and negative signals with visible feedback loops regulating MPF kinase (Fig 1B) and APC/C ubiquitin ligase (Fig 1C) in the overall mitotic biopathway. Also unique to this model, the mitotic interacting reactions have been assembled in a modular

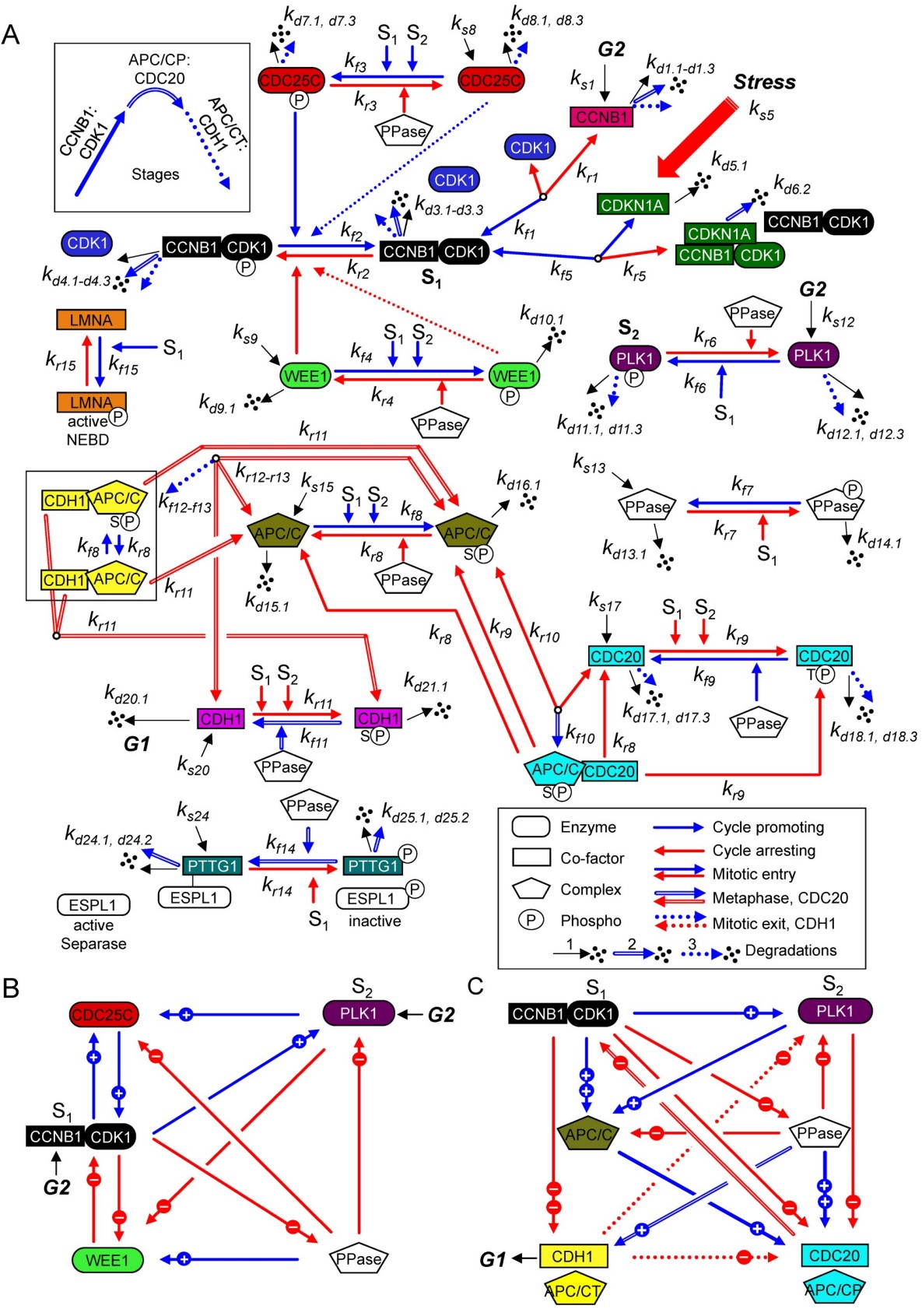

**Fig 1. Regulation network of the mitotic cell cycle in human cells.** The mitotic biopathway for the reaction network of key mitotic proteins is converted into a mathematical structure based on ODEs involving the law of mass conservations and a hybrid framework combining mass action and Michaelis-Menten kinetics. **(A)** We have divided the biopathway into three stages (see inset) defined by arrow styles. Each factor, as defined in the text and Supplemental Information, is identified by a unique color conserved throughout the article. Major stages include: (1) Mitotic entry (solid arrows) involving CCNB1, CDK1, preMPF (CCNB1:CDK1P), MPF (CCNB1:CDK1), PLK1, WEE1, CDC25C, PPase (phosphatase), and LMNA, which is a substrate of MPF kinase (P, phosphorylated; NEBD, nuclear envelop breakdown); (2) Metaphase to anaphase transition (double arrows) involving APC/C:CDC20, PPase, and PTTG1 (Securin) in complex with ESPL1 (Separase); and (3) Mitotic exit (dashed arrows) involving APC/CT:CDH1 (T, total). Here, S1 denotes active MPF kinase (CCNB1:CDK1) and S2 denotes active PLK1 kinase (phosphorylated), which are major signals for many mitotic cell cycle reactions, as indicated. Reactions involving Cyclin-dependent kinase inhibitor CDKN1A (p21$^{CIP1}$) have been included, and locations for G2 signals and exit into G1 are noted. The model consists of 26 ODEs governing 15 interaction reactions ($k_{fn}$ forward rate constant and $k_{rn}$ reverse rate constant for reaction $n$) of 12 key mitotic proteins and associated protein complexes and includes synthesis ($k_s$: synthesis rate constant) and multiple degradations ($k_{dn.1}$: self-degradation rate constant, $k_{dn.2}$: degradation rate constant by APCP/CDC20, and $k_{dn.3}$: degradation rate constant by APC/CT:CDH1). **(B, C)** Simplified wiring diagrams based on (A), which identify both positive (blue, plus symbol) and negative (red, minus symbol) signals regulating the MPF kinase (CCNB1:CDK1) (B), and APC/CP:CDC20 and APC/CT:CDH1 ubiquitin ligase complexes with several signal strengths (C).

format allowing for future refinement and expansion of mechanistic details such as inclusion of MPF regulation by hGWL kinase [10, 27, 30, 32, 35, 36].

We have centered the process around the formation of CCNB1:CDK1 complex which we refer to as either inactive preMPF (CCNB1:CDK1P) or active MPF kinase (CCNB1:CDK1) (Fig 1A and 1B). Inactive preMPF (CCNB1:CDK1P) is phosphorylated at tyrosine 15 (Y15). Numerous mechanisms influence preMPF and MPF formation and we have focused on a subset of essential reactions in constructing this new model (reviewed in [2]). These include the dynamics of CCNB1 levels and binding to CDK1, activity of PLK1 kinase, and changes in phosphorylation of CDK1. Simulating the biopathway dynamics via ODEs over one cycle (Fig 2) shows the complexity of relationships and how they change over time. We have defined factors using unique colors and line patterns conserved throughout the article as described in Fig 2. For example, the level of total CCNB1 (CCNB1T) (Fig 2A, magenta) at any given time is determined by the rates of synthesis and degradations with CCNB1T equal to the sum of free CCNB1 and CCNB1 complexed with CDK1 (Fig 2A, blue). For the purposes of this work, we have maintained constant levels of CDK1T and have not accounted for cellular compartmentalization. Accumulation of preMPF and active MPF kinase (Fig 2A, black) is associated with a concurrent reduction in free CDK1. Subsequently, the conversion of preMPF to MPF kinase (Fig 2A, black) requires the loss of phosphorylation at Y15 which is tightly regulated. We have accounted for key reactions in this process. Expression of WEE1 kinase (Fig 2B, green) promotes the accumulation of phosphorylated preMPF. The conversion of preMPF to active MPF kinase occurs upon activation of CDC25C phosphatase (Fig 2B, red) and loss of WEE1 kinase. At the initiation of mitosis, PLK1 kinase (Fig 2B, purple) contributes to increasing CDC25CP activation and, along with accumulating MPF, converts WEE1 to WEE1P. Phosphorylation of WEE1 stimulates its loss by degradation mediated through E3 ligase SCF-βTrCP [41]. The combination of these reactions results in maximal MPF kinase activity and phosphorylation of substrates including LMNA (Fig 2A, orange). This initiates nuclear envelop breakdown (NEBD) (Fig 1A).

Active MPF kinase (CCNB1:CDK1) phosphorylates diverse substrates during mitosis [42]. In addition to CDC25C and WEE1, we have included the substrates PLK1, LMNA, APC/C, CDC20, CDH1, PTTG1 (Securin) and PPases (phosphatases). Phosphorylation of APC/C (Fig 2C, olive green) promotes the transition through mitosis involving sequential activation of APC/CP:CDC20 (Fig 2C, light blue) followed by APC/CT:CDH1 (Fig 2D, yellow). This transition is regulated by differential amino acid substrate preferences by MPF kinase and PP2A phosphatase as well as substrate preferences for ubiquitination by APC/C [13, 43]. Activation of APC/CP:CDC20 requires dephosphorylation of CDC20P (Fig 2C, light blue) which

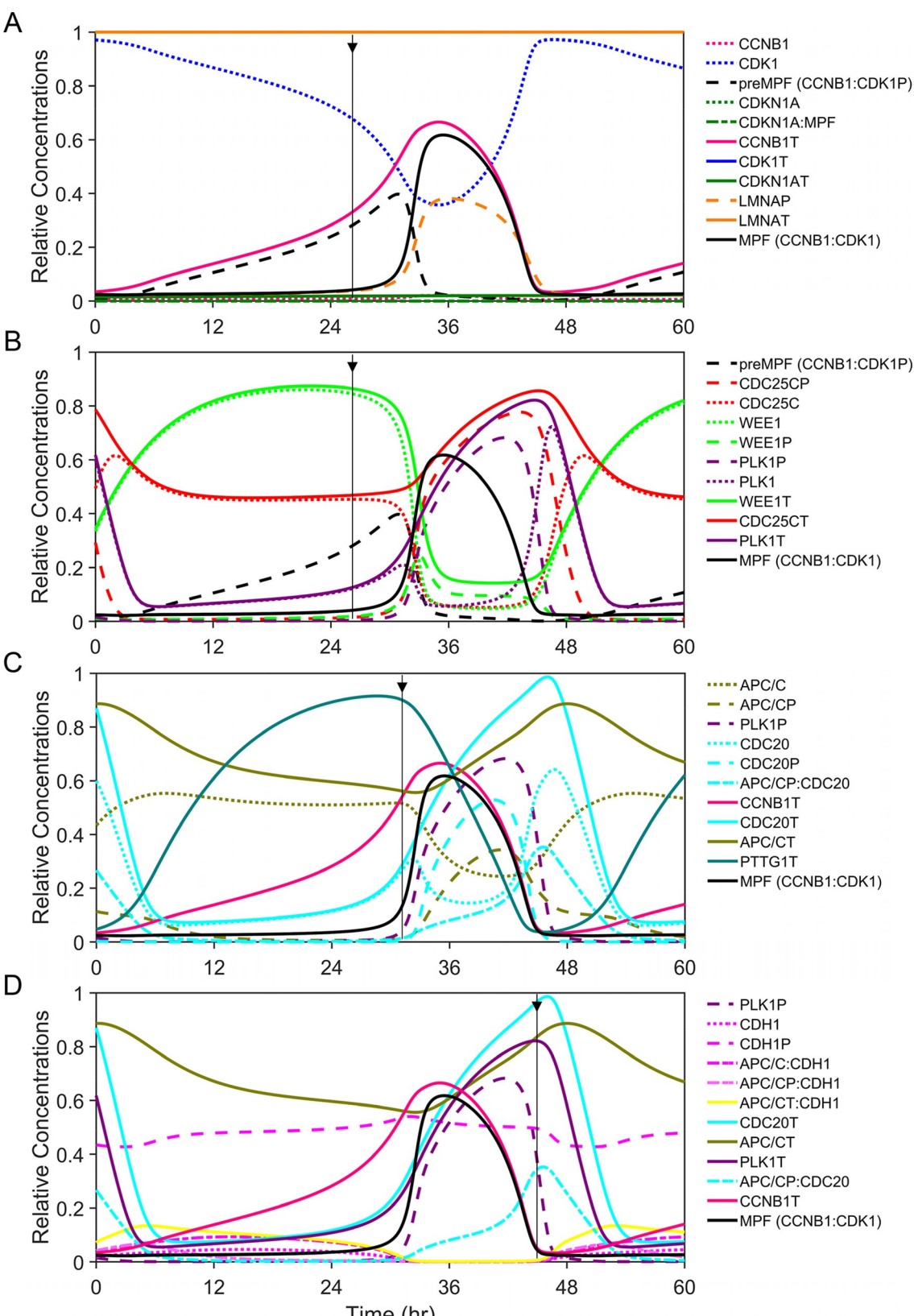

**Fig 2. Subsets of regulatory relationships and functional outcomes driving the mitotic cell cycle.** Simulations of relative concentrations for key mitotic proteins and protein complexes during one cycle of mitosis over a 48 hr time scale, centered on activation/inactivation of MPF kinase (CCNB1:CDK1, black line). Each factor is identified by a unique color for line and bar graphs throughout the article. Oscillations of factors are shown relative to each and include total protein or protein complexes (T, solid lines), phosphorylated proteins (P, wide dashed lines), free unphosphorylated proteins (narrow dashed lines), and regulated interactions (mixed dashed lines). Additional phases of the cell cycle are not considered in this new model. Biochemical reactions, ODEs, and optimized model parameter values are presented in the Supporting Information. **(A)** Simulating increasing levels of CCNB1T (magenta) in the presence of constant CDK1T (blue) results in formation of preMPF (CCNB1:CDK1P, black) then active MPF kinase (CCNB1:CDK1, black). The vertical arrow approximates the onset of active MPF kinase (black). Subsequent changes include reduced free CDK1 (blue) and increased LMNA phosphorylation (LMNAP, orange). These occur in the absence of CDKN1AT (p21$^{CIP1}$, dark green) expression. Concentrations of LMNAT (orange) and CDK1T (not visible) are constant and equal to 1. **(B)** Simulating increasing PLK1T (purple) and active PLK1P kinase (purple) results in active CDC25CP phosphatase (red) and inactivation WEE1P kinase (green) with its subsequent degradation and accompanying activation of MPF. The vertical arrow again approximates the onset of active MPF kinase (black). **(C)** Simulating the transition to metaphase by PLK1P- (purple) and MPF-mediated phosphorylation of APC/C (APC/CP, olive green) and CDC20 (CDC20P, light blue) followed by rapid dephosphorylation of CDC20P (light blue) and activation of the APC/CP:CDC20 (light blue) ubiquitin ligase complex. The vertical arrow approximates the onset of active APC/CP:CDC20 (light blue). Activation results in degradations of PTTG1T (cyan) and CCNB1T (magenta). **(D)** Simulating mitotic exit upon dephosphorylation of CDH1P (pink), formation and activation of APC/CT:CDH1 (APC/C:CDH1 and APC/CP:CDH1, yellow) ubiquitin ligase. The vertical arrow approximates the onset of active APC/CT:CDH1 (yellow), resulting in degradations of PLK1T (purple), CDC20T (light blue) and any remaining CCNB1T (magenta).

contributes to the metaphase to anaphase transition. Substrates of APC/CP:CDC20 include PTTG1T (Fig 2C, cyan) and CCNB1T (magenta). The subsequent increase in unphosphorylated CDH1 (Fig 2D, pink) results in APC/CT:CDH1 formation and activation. This ubiquitin ligase complex promotes mitotic exit by targeting CDC20T, PLK1T and remaining CCNB1T for degradation. In parameterizing this new model, values of many of these parameters are directly adopted or fine-tuned from relevant published models in the literature from other laboratories [16, 19, 20, 23–26, 44, 45] (specifically [20] related to APC/C and CDC20 interactions), and the remaining parameter values are estimated based on scientific reasoning and thermodynamic and kinetic constraints. This approach enabled us to reproduce the essential features of the human mitotic cell cycle observed experimentally. The parameter estimation approach is described in detail within S4 Appendix. We are using a hypothetical 48 hr cycle (oscillation period) to allow for visualization of key changes in the mitotic proteins through the simulations during the cell entering into M phase from late G2 phase and exiting from M phase into early G1 phase. In general, cell doubling times vary by cell type, passage number, and culture conditions. In this model, the mitotic cycle duration (i.e. the duration for the cell to enter into M phase from late G2 phase and exit from M phase into early G1 phase) is controlled by one global scaling parameter ($\alpha$), as described in S3 Appendix. Increasing the current scaling factor of 1.4 to 2.8 reduces the mitotic cycle time to 24 hr from 48 hr (S5 Appendix). In the subsequent sections, we have further tested the model to reproduce the cardinal features of human mitosis experimentally determined and published by several leading laboratories in the field of mitosis biology.

## Disruption of the CDC25C-WEE1-regulated bistable switch results in mitotic collapse

Active MPF kinase (CCNB1:CDK1) promotes mitosis. We have defined a subset of relationships focusing on changes over time in CCNB1T expression, formation of preMPF (CCNB1:CDK1P) and activation of MPF kinase, including regulators WEE1T and CDC25CT (Fig 1A and 1B). These relationships have been experimentally demonstrated *in vitro* by numerous laboratories including studies by Ovejero *et al.* [8]. Using synchronized human cells, the approximated signal intensities from published data show increasing CCNB1T in M phase (Fig 3A) accompanied by decreases in WEE1T and CDK1P (i.e. preMPF) [8]. WEE1T kinase activity is

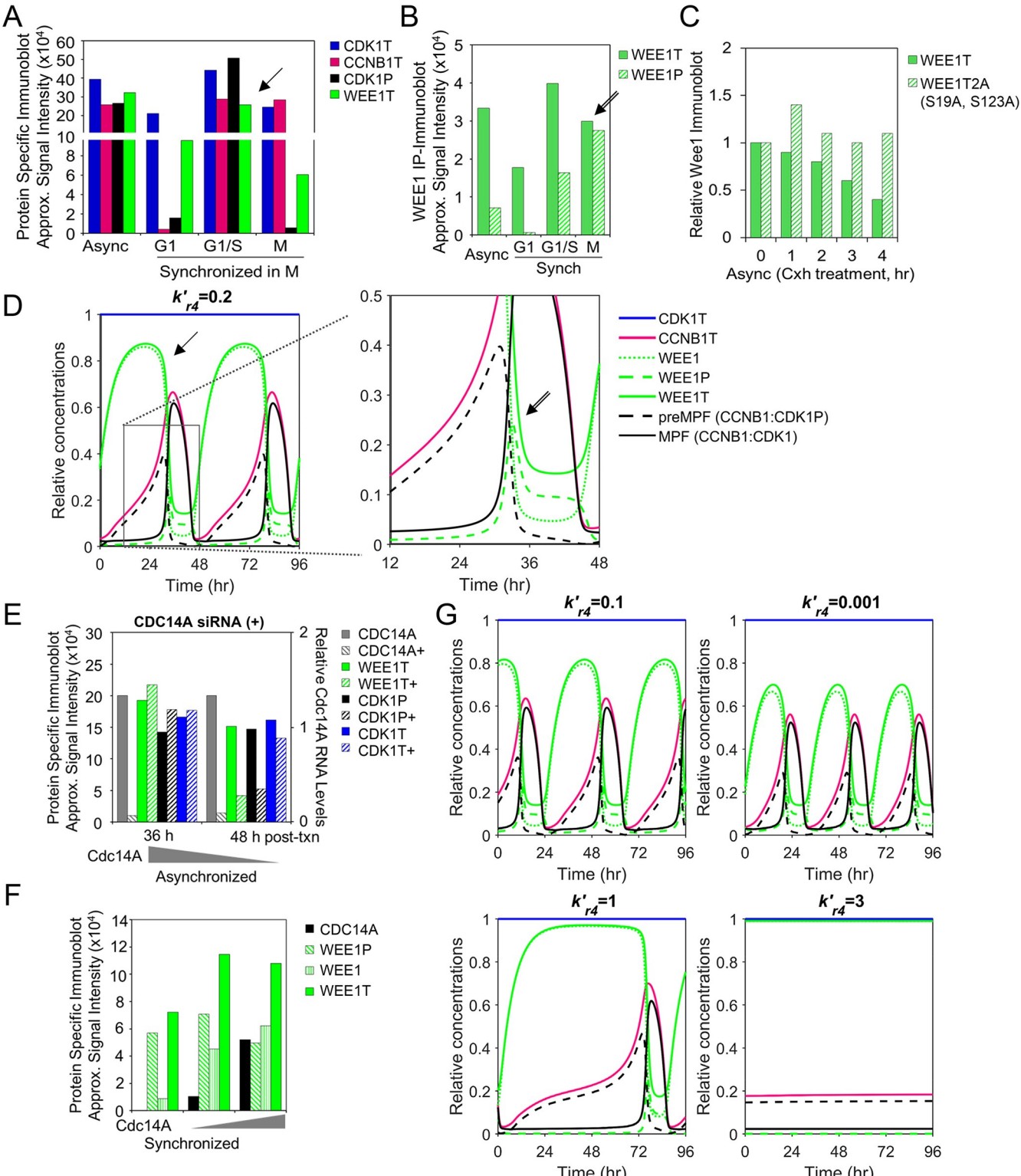

**Fig 3. WEE1-mediated regulation of preMPF and MPF oscillations.** **(A)** Approximated steady-state protein levels from studies by Ovejero *et al.* [8] using immunoblot analysis for CDK1T, CCNB1T, WEE1T and CDK1P Y15 (i.e. preMPF) in WEE1-expressing U2OS cells following release from synchronization in M phase. Arrows mark points of comparison between *in vitro* and *in silico* data. (P, phosphorylated protein; T, total protein) **(B)** Levels of WEE1P (S139) observed in immunoprecipitated WEE1T from studies by Ovejero *et al.* [8] as described in (A). **(C)** Approximated changes in protein levels for WEE1T and mutated unphosphorylated WEE1T 2A (S139A, S123A) during cycloheximide treatment by Ovejero *et al.* [8]. **(D)** Simulation of phosphatase activity (PPase,

$k'_{r4}$) in regulating free WEE1, WEE1P and WEE1T with associated oscillations in CDK1T, CCNBT, preMPF (CCNB1:CDK1P), and active MPF kinase (CCNB1:CDK1) over the indicated time periods. Boxed area is magnified. **(E)** Disrupting relationships by siRNA-mediated reduction of phosphatase CDC14A (+) in asynchronized U2OS cells and associated changes in levels of WEE1T and CDK1P Y15 (i.e. preMPF) compared to control by Ovejero *et al.* [8]. **(F)** Approximated changes in WEE1, WEE1T and WEE1P levels upon increasing expression of CDC14A phosphatase as demonstrated by Ovejero *et al.* [8]. **(G)** Simulated decrease in phosphatase (PPase, i.e. CDC14A) activity ($k'_{r4}$) and associated changes in protein concentrations and mitotic oscillations of CDK1T, CCNBT, WEE1, WEE1T, WEE1P, preMPF (CCNB1:CDK1P), and active MPF kinase (CCNB1:CDK1). In contrast, simulated increase in PPase activity ($k'_{r4}$) and associated changes resulting in a mitotic collapse with no active MPF kinase (CCNB1:CDK1) over 96 hr.

regulated by changes in posttranslational modifications with increasing phosphorylation resulting in proteasome-dependent degradation of WEE1P [8]. Immunoprecipitated WEE1T contains increasing WEE1P at M phase (Fig 3B), and WEE1P is degraded compared to a non-phosphorylatable WEE1 mutant (Fig 3C) [8]. We have captured these features in the simulations of our computational model. Increasing concentration of CCNB1T over time in the presence of WEE1T kinase and stable CDK1T levels is accompanied by increasing preMPF formation (Fig 3D). We have identified several areas exhibiting similar qualitative changes between *in vitro* and *in silico* data using arrows here (Fig 3A and 3D, arrow) and in other figures throughout the article. As WEE1T concentrations decrease due to WEE1P formation and degradation (Fig 3B and 3D, double arrow), we observe decreasing preMPF and increasing active MPF kinase.

Dephosphorylation of WEE1P to WEE1 is mediated by phosphatases (PPases) including CDC14C [8]. These changes occur in our model simulations and can be seen at 36 hr exhibiting a decrease in WEE1T upon increasing WEE1P (Fig 3B and 3D, double arrow). We have tested our model by comparing published experimental perturbations designed to dissect the system to perturbations *in silico*. Experimental depletion of CDC14A (Fig 4E, plus symbol) results in relative decreases in both WEE1T and CDK1P (i.e. preMPF) with little impact on CDK1T [8], suggesting an acceleration into mitosis. In contrast, experimentally overexpressing CDC14A (Fig 3F) results in increased stead state levels of WEE1T [8]. We simulated the experiments by reducing the rate constant $k'_{r4}$ (Fig 1A) from 0.2 to 0.1 and 0.001. These changes result in reduced maximal WEE1T levels and increased frequency of oscillations (Fig 3G). Upon simulating an increase in $k'_{r4}$ from 0.2 to 1 and 3, we observed increased WEE1T and reduced frequency of oscillations, ultimately ending in a mitotic collapse with no active MPF kinase (CCNB1:CDK1) (Fig 3G). These comparisons suggest that the model defining WEE1T and regulation of MPF kinase *in silico* exhibit similar dynamic relationships to those observed from the *in vitro* studies [8].

CDC25C phosphatase promotes activation of MPF kinase (CCNB1:CDK1) by dephosphorylating preMPF (CCNB1:CDK1P). Active MPF kinase accelerates the process by phosphorylating both CDC25C (an activating event) and WEE1 (an inhibiting event) creating a bistable switch (Fig 1A and 1B). This has been experimentally and computationally demonstrated by Mochida *et al.* [37]. Active MPF kinase phosphorylates additional substrates including Histone H3, LMNA and APC/C subunits APC1 and APC3 [42]. Studies completed by Potapova *et al.* [9] demonstrate a relative increase in CCNB1T accompanied by decrease in CDK1P (i.e. preMPF) at constant CDK1T levels following the release of synchronized human cells (Fig 4A). These changes are accompanied by an increase in APC/CP and phosphorylated H3 at S10 (Fig 4A). Data from Potapova *et al.* [9] was used by Tuck *et al.* [27] to uncover a latching-switch mechanism controlling the G2/M transition during a single cycle. Here, we have tested our model during multiple cycles to reproduce the behaviors experimentally demonstrated by Potapova *et al.* [9]. We have defined a subset of these relationships including CDC25CP-mediated activation of MPF kinase and subsequent MPF-mediated phosphorylation of CDC25C, WEE1, APC/C and LMNA (Fig 1A). As seen in model simulations, increasing concentrations

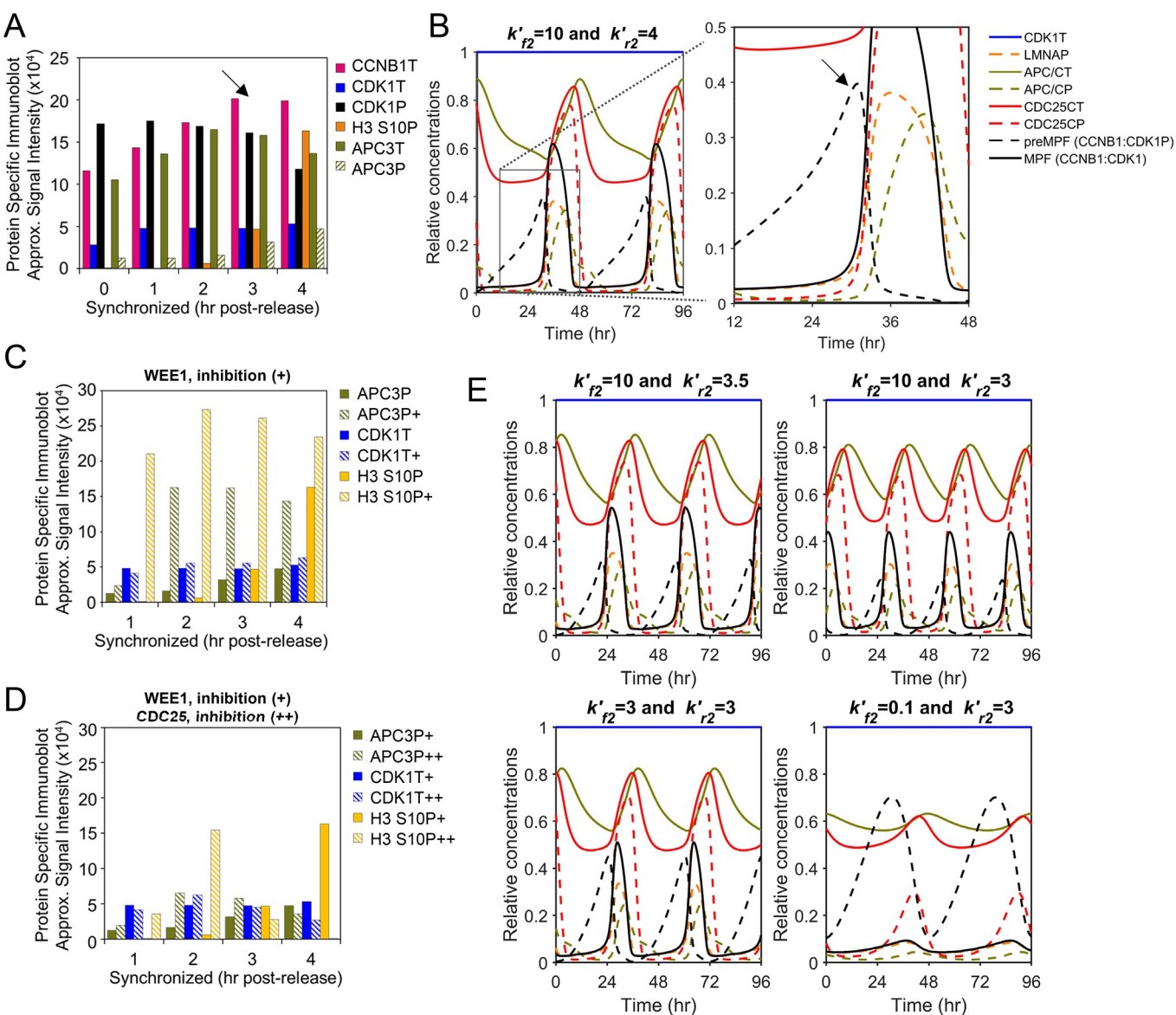

**Fig 4. WEE1 and CDC25C regulation of preMPF and activation of MPF kinase. (A)** Approximated steady-state protein levels from studies by Potapova *et al.* [9] using immunoblot analysis for CCNB1T, CDK1T, CDK1P (i.e. preMPF), phosphorylated H3 S10P, APC3 (i.e. APC/CT), APC3P (i.e. APC/CP) in HeLa cells following release from synchronization in the S/G2 phase. Arrows mark points of comparison between *in vitro* and *in silico* data. (P, phosphorylated protein; T, total protein) **(B)** Simulation of WEE1 kinase ($k'_{r2}$) and CDC25C phosphatase ($k'_{f2}$) activities in regulating oscillations in CDK1T, LMNAP, APC/CT, APC/CP, CDC25CT, CDC25CP, preMPF (CCNB1:CDK1P), and active MPF kinase (CCNB1:CDK1) over the indicated time periods. Boxed area is magnified. **(C)** Approximated steady-state protein levels for APC3P, CDK1T, and H3 S10P from studies by Potapova *et al.* [9] involving chemical inhibition (+) of WEE1 kinase activity using synchronized HeLa cells compared to control. **(D)** Inhibition of WEE1 with inhibition of CDC25 (+) activities in synchronized HeLa cells by Potapova *et al.* [9] and associated changes in APC3P, CDK1T, and H3 S10P resulting in a mitotic collapse. **(E)** Simulated inhibition of WEE1 kinase ($k'_{r2}$) and associated changes oscillations in CDK1T, LMNAP, APC/CT, APC/CP, CDC25CT, CDC25CP, preMPF (CCNB1:CDK1P), and active MPF kinase (CCNB1:CDK1) over the indicated time period. Also, simulated inhibition of both WEE1 ($k'_{r2}$) and CDC25C ($k'_{f2}$) and associated changes resulting in a mitotic collapse.

of MPF kinase over time is accompanied by decreasing preMPF and increasing CDC25CP, LMNAP and APC/CP (Fig 4A and 4B, arrow). Future refinements of our present model will be completed to distinguish between nuclear and cytoplasmic localization in regulation which plays an important role in these processes.

To test CDC25C and WEE1 regulation of active MPF kinase (CCNB1:CDK1), we again compared experimental data to model simulations. To disrupt these relationships, Potapova *et al.* [9] inhibited WEE1 kinase activity (Fig 4C, plus symbol) in synchronized human cells, demonstrating rapid increases in phosphorylated APC3P (i.e. APC/CP) and H3 S10P compared to control with little impact on CDK1T. These observations indicate an acceleration into mitosis also noted by Tuck *et al.* [27]. To test our model, we simulate WEE1 inhibition by reducing rate constant $k'_{r2}$ (Fig 1A) from 4 to 3.5 and 3. This resulted in accelerated oscillations including changes in APC/CP and LMNAP compared to normal conditions (Fig 4E). Potapova *et al.* [9] also demonstrated a mitotic collapse upon inhibiting both WEE1 and CDC25 (Fig 4D, plus symbol). To simulate these conditions, we set $k'_{r2}$ to 3 and decreased CDC25C activities by reducing $k'_{f2}$ (Fig 1A) from 10 to 3 and 0.1 (Fig 4E). This ended in a mitotic collapse defined by negligible MPF kinase activity and substrate phosphorylation which is consistent with the experimental studies [9]. Together, our model simulates activities and outcomes observed from *in vitro* studies on CDC25C and WEE1 regulation MPF kinase activity.

## PLK1 is the dominant regulator of mitosis and mitotic entry

The PLK1 kinase is activated in G2 and is a key regulator of mitotic entry. We have accounted for the activities of PLK1 and PLK1P in regulating activation of MPF kinase (CCNB1:CDK1) (Fig 1A). This is an additional unique feature of our new computational model of the mitotic cell cycle. Active PLK1P initiates mitotic entry by phosphorylating CDC25C and WEE1. Using synchronized human cultured cells, Watanabe *et al.* [38] demonstrated that increasing levels of PLK1T is accompanied by increasing WEE1P and decreasing WEE1T (Fig 5A). Studies by Gheghiani *et al.* [7] demonstrated increasing PLK1P is accompanied by increasing CDC25CP and decreasing CDK1P (i.e. preMPF) (Fig 5B). In simulating these relations, we observe increasing levels of PLK1T and, to a lesser extent, PLK1P prior to the accumulation of CDC25CP and reductions in preMPF and WEE1P (Fig 5A–5C, arrows). The magnitude of PLK1P accumulation is limited in this base model due to the absence of G2 signals [7, 46]. Our model simulations follow the qualitative changes over time involving PLK1 as observed from *in vitro* studies [7, 38].

We have further validated the model by comparing *in vitro* data to simulated data following inhibition of WEE1 kinase and PLK1 kinase. Chemical inhibition of WEE1 using synchronized human cells results in decreased CDK1P (i.e. preMPF), increased PLK1P and CDC25CP, and premature entry into mitosis as shown in studies by Gheghiani *et al.* [7] (Fig 5D). In contrast, Gheghiani *et al.* [7] also demonstrated that chemical inhibition of PLK1 delays entry into mitosis which is negated upon overexpression of a phosphomimic of CDC25C, CDC25C5E (i.e. active CDC25CP phosphatase) (Fig 5E). In our model, inhibition of WEE1 kinase *in silico* by reducing $k'_{r2}$ (Fig 1A) from 4 to 3.5 and 3 results in increased frequency of mitotic oscillations including premature increases in PLK1P and CDC25CP and decrease in preMPF (CCNB1:CDK1P) (Fig 5F). To simulate inhibition of PLK1, we increased the rate constant $k'_{r6}$ (Fig 1A) from 0.5 to 1.5 reducing PLK1P which is the active form of PLK1 kinase. This inhibition extended the length of the mitotic cell cycle (Fig 5G). To simulate a phosphomimic of CDC25C, we increased the rate of CDC25C phosphorylation at $k'_{f3}$ (Fig 1A) from 1 to 1.5 and 2 (Fig 5G). This change restored oscillations under PLK1 inhibition ($k'_{r6}$ = 1.5) and is consistent with studies by Gheghiani *et al.* [7]. Overall, these simulations suggest that our formulated reactions and ODEs capture the dynamic and experimentally-defined relationships observed between PLK1 and regulators of active MPF kinase (CCNB1:CDK1).

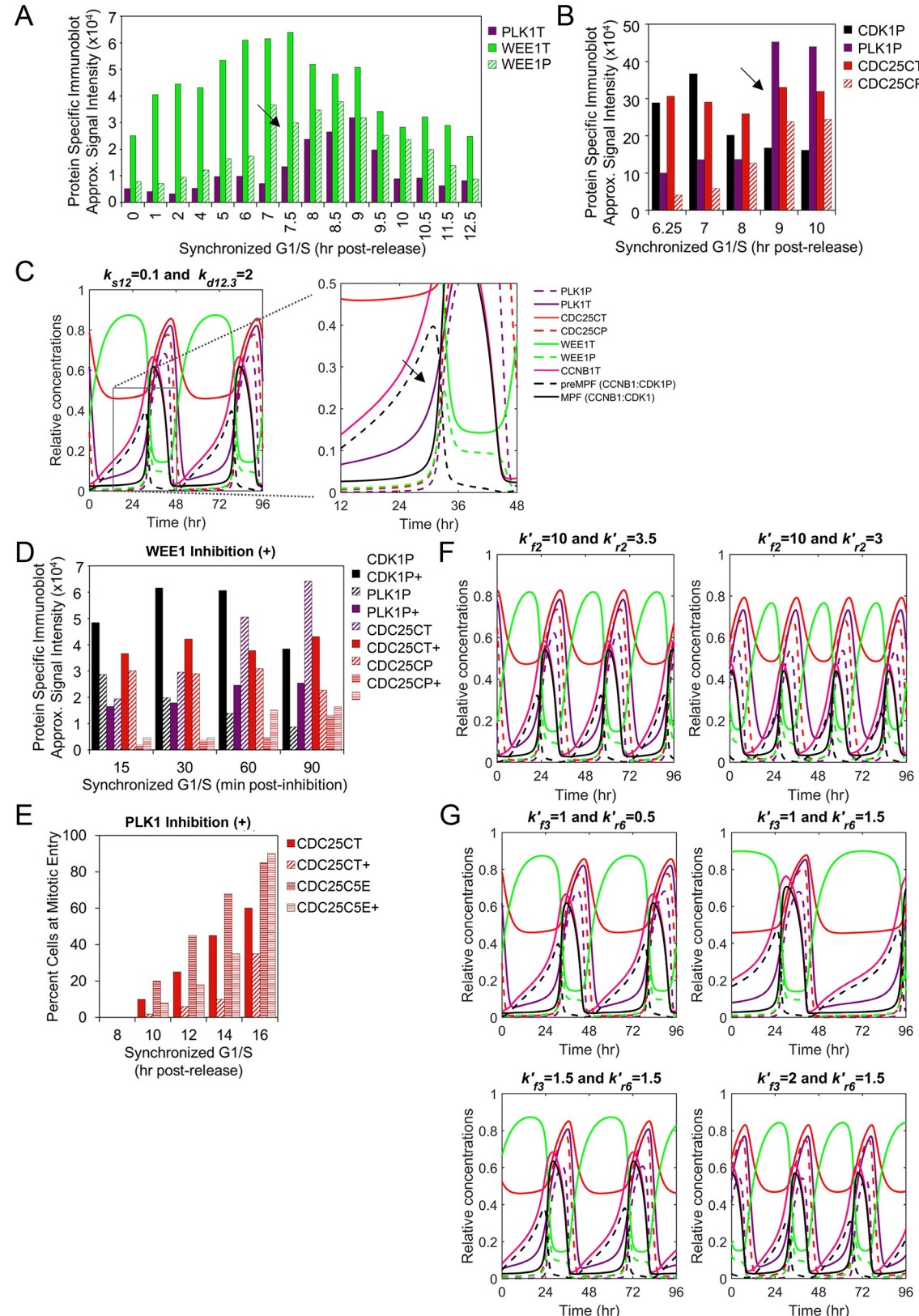

**Fig 5. Role of PLK1 in stimulating mitotic entry and MPF kinase activation. (A)** Approximated steady-state protein levels from studies by Watanabe *et al.* [38] using immunoblot analysis of PLK1T, WEE1T and WEE1P in HeLa cells following release from

synchronization in the G1/S phase with mitosis observed between 9–10 hr. Arrows mark points of comparison between *in vitro* and *in silico* data. (P, phosphorylated protein; T, total protein) **(B)** Approximated steady-state protein levels of CDK1P (i.e. preMPF), PLK1P, CDC25CT and CDC25CP following release of HeLa cells from G1/S from studies by Gheghiani *et al.* [7]. **(C)** Simulation of PLK1 expression with oscillations in PLK1P, PLK1T, CDC25CT, CDC25CP, WEE1T, WEE1P, CCNBT, preMPF (CCNB1:CDK1P) and active MPF kinase (CCNB1:CDK1). Boxed area is magnified. **(D)** Approximated steady-state protein levels from experiments by Gheghiani *et al.* [7] described in (B) upon inhibition (+) of WEE1 kinase activity relative to control. **(E)** Percent of cells at mitotic entry as determined by Gheghiani *et al.* [7] using HEK293 cells expressing a wild-type CDC25C reporter or a phosphomimic CDC25C5E and following inhibition (+) of PLK1 compared to control. **(F)** Simulated inhibition of WEE1 ($k'_{r2}$) and associated changes in factors as described in (C). **(G)** Simulated inhibition of PLK1 ($k'_{r6}$) for wild-type CDC25C or phosphomimic CDC25C5E and associated changes as described in (C). Phosphomimic CDC25C5E is introduced by reducing PPase activity at $k'_{f3}$ resulting in increased CDC25CP.

## High threshold requirements for a temporary CDKN1A (p21$^{CIP1}$)-mediated mitotic arrest

The CDK inhibitor, CDKN1A (p21$^{CIP1}$) binds to multiple CDKs and Cyclin subunits, regulating cell cycle and mediating arrests at multiple phases. CDKN1A has a short half-life with steady state levels determined by rates of synthesis and degradations involving E3 ubiquitin ligases. Following induction, CDKN1A binds to both CDK1 and CCNB1 and has been demonstrated to inhibit mitosis [47]. Degradation of CDKN1A during mitosis involves APC/CP: CDC20 [48]. We have accounted for these reactions in our new integrated computational model of mitosis (Fig 6A). *In vitro* kinase assays by Harper *et al.* [49] have demonstrated that CDKN1A inhibits MPF (CCNB1:CDK1)-mediated phosphorylation of several substrates (Fig 6B). However, this inhibition requires higher concentration of CDKN1A when compared to interactions with other CDK complexes including Cyclin A:CDK2 [49]. The kinetics of inhibition suggest that three molecules of CDKN1A are required to inhibit MPF kinase (Fig 6B). We have incorporated CDKN1A into the model using these quantitative data [41]. To simulate CDKN1A activities, we have increased CDKN1A synthesis by altering $k_{s5}$ (Fig 6C) from 0.0001 to 0.1 and 1.5 which results in increased CDKN1A:MPF association, reduced LMNAP levels and an eventual mitotic collapse (Fig 6C). We can restore oscillations to similar levels of active MPF kinase by increasing the rate of CDKN1A degradation by APC/CP:CDC20 using $k_{d6.2}$ from 1 to 10 (Fig 6C). However, oscillations exhibit an extended mitotic phase compared to normal oscillations. Alternatively, normal oscillations can be restored by reducing the rate of CDKN1A:MPF assembly, $k_{r5}$ (Fig 6A) from 80 to 1 under conditions of high synthesis ($k_{s5}$ = 0.2) (Fig 6D). CDKN1A is observed to be expressed at low levels in human cells, and cells deficient in CDKN1A exhibit defects in the mitotic cell cycle, indicating the requirement for a basal level of expression. In our model, we have determined that $k_{s5}$ can range from 0.0001 to 0.01 before MPF levels drop by more than 10% (Fig 6E) with a mitotic arrest starting when $k_{s5}$ > 0.1 (Fig 6C). With the addition of CDKN1A to our model, we have accounted for the activities of a critical cell cycle regulator allowing for future model expansion to include of cellular stress pathways associated with dysregulations in cancer and viral infection.

## The model predicts differential substrate preferences between MPF kinase and PP2A phosphatase

The transition through mitosis requires activation of the APC/CP:CDC20 ubiquitin ligase, inactivation of MPF kinase (CCNB1:CDK1), and mitotic exit upon activation of APC/CT: CDH1 ubiquitin ligase. This transition is coordinated by the different amino acid substrate preferences between MPF kinase (serine>threonine) and PP2A (PPase) phosphatase (phosphothreonine>phosphoserine) [13]. We have accounted for the activities of MPF and PP2A in regulating APC/C and its coactivators, CDC20 and CDH1 (Fig 1A and 1C). Activation of

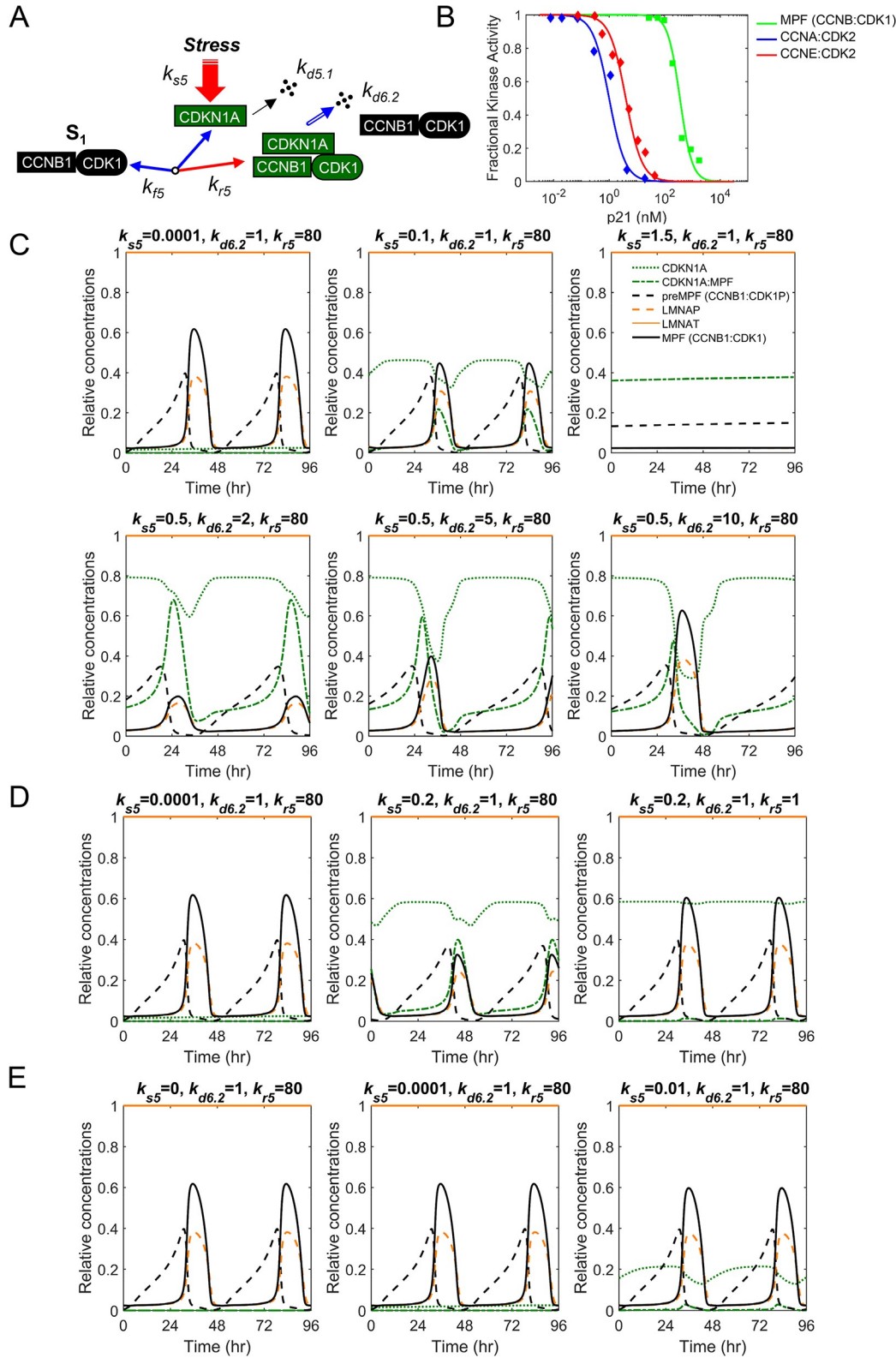

**Fig 6. Regulation of the mitotic cell cycle by cyclin-dependent kinase inhibitor, CDKN1A (p21$^{CIP1}$). (A)** Mechanisms regulating CDKN1A steady-state levels including synthesis, degradations, and association with MPF kinase (CCNB1: CDK1). **(B)** *In vitro* kinase activities as determined by Harper *et al.* [49] for CDKN1A inhibition of MPF (CCNB1:CDK1), Cyclin A:CDK2, and Cyclin E:CDK2 kinases following increasing concentrations of CDKN1A. **(C)** Simulated increase in

CDKN1A synthesis ($k_{s5}$) and associated changes in CDKN1A, CDKN1A:MPF, preMPF (CCNB:CDK1P), LMNAP, LMNAT and active MPF kinase resulting in a mitotic collapse. Also, simulated increase in CDKN1A degradation ($k_{d6.2}$) and restored oscillations. **(D)** Restoring oscillations of factors in described in (C) by disrupting the association ($k_{r5}$) between CDKN1A:MPF. **(E)** The range of CDKN1A synthesis ($k_{s5}$) with normal mitotic oscillations in factors described in (C).

APC/C requires phosphorylation by MPF and PLK1 kinases to APC/CP. This promotes binding of unphosphorylated CDC20 (APC/CP:CDC20). Using human cells synchronized in either G1/S (Fig 7A) or M (Fig 7B), Zhao *et al.* [40] demonstrate initial increases in the levels of CCNB1T, PLK1T, CDC20T and CDH1T (Fig 7A). This is followed by reductions in CCNB1T, CDC20T, PTTG1T and APC/CP and, subsequently, a reduction in PLK1T (Fig 7B) [40]. The levels of CDH1T and APC/CT are relatively stable over time (Fig 7B) [40]. In simulating these relations, we observe similar trends including increasing levels of CCNB1T, MPF kinase, PLK1T and CDC20T (Fig 7A and 7C, arrows). Upon increasing level PPase (i.e. PP2A) in the simulation, we observe decreasing levels of CCNB1T, MPF and PTTG1T. This is followed by decreases in APC/CP, CDC20T and PLK1T (Fig 7B and 7C, arrows). The simulations also show smaller changes in APC/CT and CDH1T levels which is consistent with data by Zhao *et al.* [40]. To further validated the model, we have compared additional *in vitro* data to *in silco* simulated data focusing on changes in CCNB1T levels upon disrupting CDC20. Hein *et al.* [13] replaced endogenous CDC20 with a mutant version of CDC20 containing serine in place of threonine residues. These changes make CDC20P less susceptible to dephosphorylation by PP2A phosphatase (i.e. PPase). Hein *et al.* [13] demonstrate that expression of mutated CDC20 results in sustained levels of CCNB1T in synchronized human cells (Fig 7D). To simulate reduced CDC20P dephosphorylation by PPase (i.e. PP2A), we have reduced the rate constant $k'_{f9}$ (Fig 1A) from 20 to 16. This change results in sustained CCNB1T levels and an extended mitotic phase (Fig 7E). Again, our computational model simulates the approximate activities and outcomes observed from published *in vitro* studies.

Unlike the regulation of MPF kinase (CCNB1:CDK1) involving feedback loops (Fig 1B), the timing of APC/C activation requires differential changes in phosphorylation (Fig 1C) [6]. Our model simulations show increasing MPF kinase activity mediates phosphorylation of APC/C to APC/CP (pS) while increasing PPase phosphatase activity mediates dephosphorylation of CDC20P (pT) to CDC20 (Fig 8A, arrow). These modifications result in activation of the APC/CP:CDC20 ubiquitin ligase complex. Hein *et al.* [13] have demonstrated that MPF kinase rapidly phosphorylates serine residence relative to threonine residues. In contrast, PP2A phosphatase (i.e. PPase) rapidly dephosphorylates threonine residues relative to serine residues. Therefore, for mitosis to progress, we predict that $k'_{f8}$ (MPF) $> k'_{r8}$ (PP2A) in regulating APC/C (pS), $k'_{r9}$ (MPF) $< k'_{f9}$ (PP2A) in regulating CDC20 (pT), and $k'_{r11}$ (MPF) $> k'_{f11}$ (PP2A) for regulating CDH1 (pS) (Fig 8B). More importantly, $k'_{f9}$ for PP2A-mediated dephosphorylation of threonine should be much greater than $k'_{r8}$ and $k'_{f11}$ for PP2A-mediated dephosphorylation of serine. To test these requirements, we varied the rate constants controlled by PP2A phosphatase activity and determined changes in MPF kinase oscillations. Upon increasing $k'_{f9}$, we observed the onset of oscillations of MPF kinase when $k'_{f9} > 16$ (Fig 8C). In contrast, upon increasing $k'_{r8}$ or $k'_{f11}$, we observe the loss of MPF oscillations when $k'_{r8} > 0.08$ and $k'_{f11} > 0.02$, respectively (Fig 8D). These results confirm our predictions and validate the experimentally-determined mechanism by Hein *et al.* [13] required to drive mitotic progression.

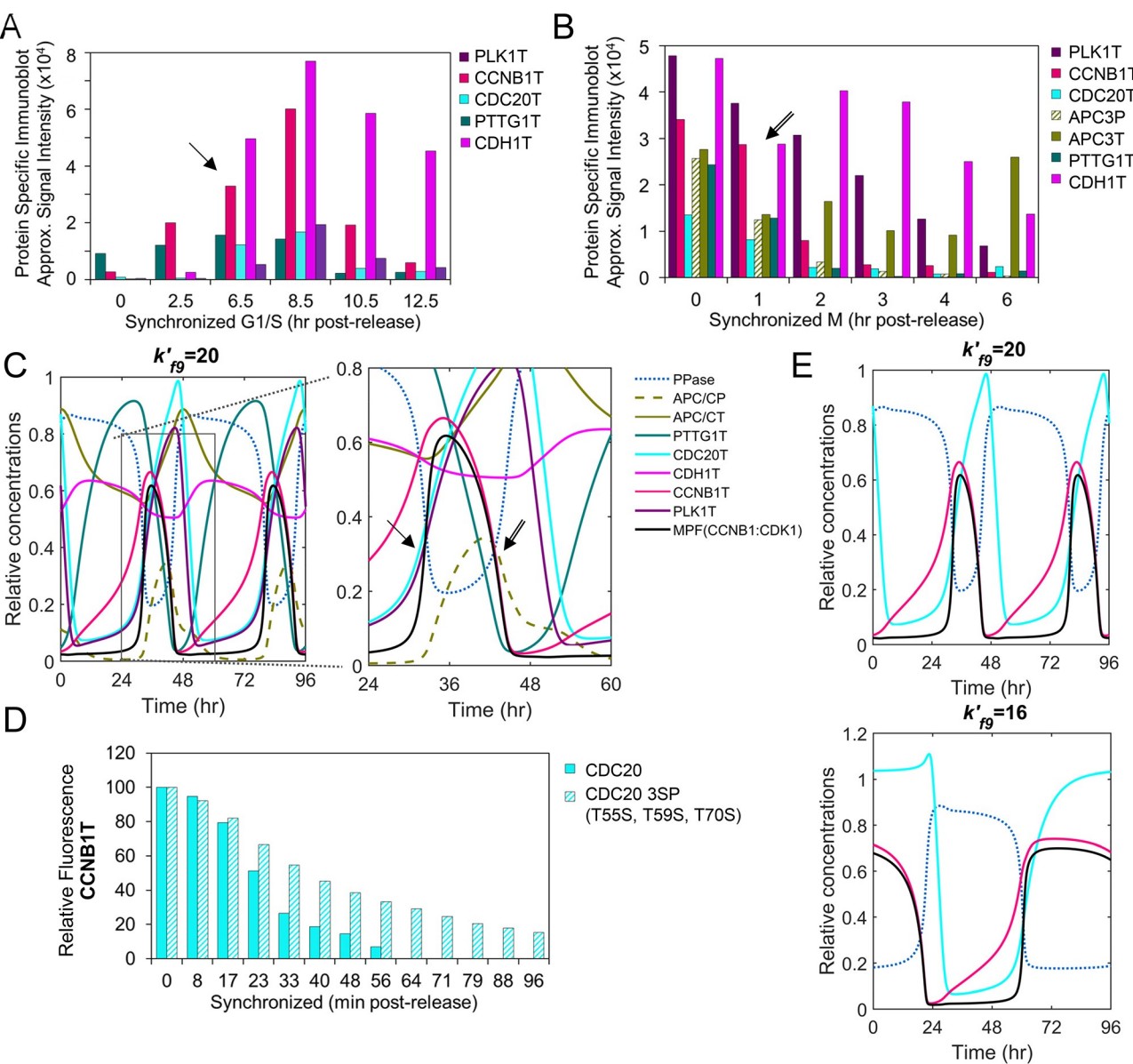

**Fig 7. Activation of APC/C and substrate degradations. (A)** Approximated steady-state protein levels from studies by Zhao *et al.* [40] using immunoblot analysis of PLK1T, CCNB1T, CDC20T CDH1T and PTTG1T in HeLa cells following release from synchronization in G1/S. Arrows mark points of comparison between *in vitro* and *in silico* data. (P, phosphorylated protein; T, total protein) **(B)** Immunoblot analysis as described in (A) by Zhao *et al.* [33] and including APC3T (i.e. APC/CT) and APC3P (i.e. APC/CP) but following release from M phase. **(C)** Simulation of APC/C ubiquitin ligase regulation including changes in PPase, APC/CP, APC/CT, PTTG1T, CDC20T, CDH1T, CCNB1T, PLK1T and MPF kinase (CCNB1:CDK1). Boxed area is magnified. (P, phosphorylated protein; T, total protein). **(D)** Kinetics of CCNB1T levels from studies by Hein *et al.* [13] using HeLa cells expressing wild-type CDC20 or mutated CDC20 3SP (T55S, T59S, T70S) following release from synchronization in G1/S. **(E)** Simulated changes over time upon reducing PPase activity ($k'_{f9}$) (i.e. PP2A) and associated changes in PPase, CDC20T, CCNB1T and MPF kinase (CCNB1:CDK1).

## Defining the contribution of APC/C degradation to mitotic collapse during herpesvirus infection

We developed this novel integrated computational model for the purpose of generating new hypotheses and predict new experiments designed to study disease. Numerous pathogenic states involve altered mitotic progression and we have applied this model to help understand

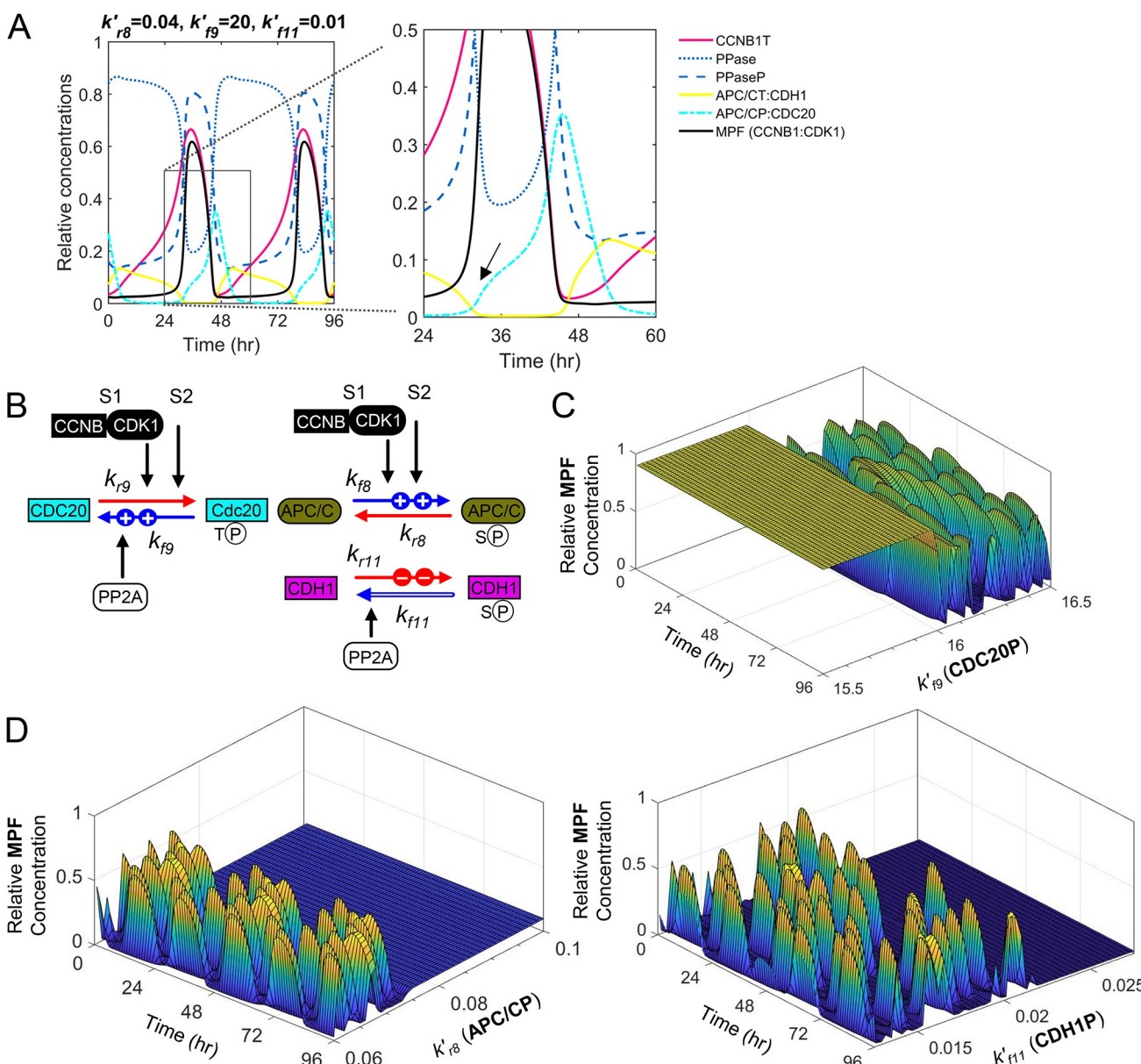

**Fig 8. Varying rates of site-specific phosphorylation and dephosphorylation drive mitotic exit. (A)** Simulation of PPase phosphatase (i.e. PP2A) and MPF kinase (CCNB1:CDK1) in regulating activation of APC/CP:CDC20 and APC/CT:CDH1 over the indicated time periods with sequential loses in CCNB1T, MPF, PPaseP, APC/CP:CDC20 and APC/CT:CDH1. Boxed area is magnified, and arrow identifies the approximate onset of APC/C:CDC20 activation. (P, phosphorylated protein; T, total protein) **(B)** MPF and PP2A substrate preferences for S (serine) and T (threonine), respectively, drive mitotic progression. This includes MPF-dependent slower phosphorylation of CDC20 ($k'_{r9}$) and faster phosphorylation of APC/C ($k'_{f8}$) and CDH1 ($k'_{r11}$), and faster PP2A-dependent dephosphorylation of CDC20P ($k'_{f9}$) and slower dephosphorylation of APC/CP ($k'_{r8}$) and CDH1P ($k'_{f11}$). **(C)** Simulated increases in PP2A-dependent dephosphorylation of CDC20P ($k'_{f9}$) result in increased oscillations of active MPF kinase occurring when $k'_{r9}$ is high. **(D)** Simulated increases in PP2A-dependent dephosphorylation of APC/CP ($k'_{r8}$) and CDH1P ($k'_{f11}$) result in loss of oscillations of active MPF kinase when $k'_{r8}$ and $k'_{f11}$ are high, respectively.

viral infections. Human cytomegalovirus (HCMV) is a ubiquitous herpesvirus and viral replication depends on cell cycle dysfunction eventually resulting in mitotic collapse [50, 51]. Late during infection, viral proteins construct a pseudo mitotic state that is necessary for efficient viral particle production [50, 51]. This state includes disruption of the nuclear lamina by

phosphorylation of LMNA by both HCMV pUL97 kinase and MPF kinase (CCNB1:CDK1) which is required for virion particle nuclear egress [51–54]. Disrupting these kinases is the defined mechanism of action for the antiviral, Maribavir [52, 54]. Entry of the infected cell into mitosis is regulated HCMV pUL21a which limits Cyclin A levels and subsequently CCNB1 levels (Fig 9A) [55]. Expressions of pUL97 and pUL21a also inactivated APC/C during infection (Fig 9B and 9C) [56–59]. However, the reason for APC/C inactivation and how it contributes to HCMV replication remain unknown.

To help answer these questions, we have used this newly developed model to simulate activities of viral proteins pUL21a and pUL97 and determine consequences to mitotic oscillations. We evaluated changes in oscillations by reducing CCNB1T at $k_{s1}$ which simulates pUL21a's impact on Cyclin A (Fig 9A) [55]. Reducing synthesis resulted in a delay in MPF kinase (CCNB1:CDK1) and an eventual mitotic collapse defined by reduced MPF and LMNAP levels and elevated APC/CT and PTTG1T levels (Fig 9D). Next, we simulated the impact of pUL97 kinase on the APC/C coactivator, CDH1 by increasing the rate constant $k'_{r11}$ (Fig 9B) [59]. This perturbation resulted in increased oscillations with an eventual mitotic collapse defined by reduced MPF, LMNAP and PTTG1T and elevated APC/CT (Fig 9E). By a different mechanism, HCMV pUL21 also mediates the degradation of subunits APC1, APC4 and APC5 resulting in the inactivation of APC/C (Fig 9C) [56, 57]. For this mechanism, we increased the rate constant $k_{d15.1}$ resulting in decreased oscillations and a mitotic collapse exhibiting elevated levels of MPF, CCNB1, PTTG1T and LMNAP (Fig 9B). These latter changes are similar to published data on HCMV infection [58, 60, 61]. Finally, we tested the impact of combining mechanisms. This includes pUL21a-mediated reduction in $k_{s1}$ and pUL97-mediated increase in $k'_{r11}$, pUL21a-mediated increase $k_{d15.1}$ and pUL97-mediated increase in $k'_{r11}$, pUL21a-mediated reduction in $k_{s1}$ and increase in $k_{d15.1}$ (APC/C) (Fig 9G), and, finally, all three mechanism combined (Fig 9H). The resulting simulations demonstrate that increased levels of LMNAP is dependent on degradation of APC/C ($k_{d15.1}$) (Fig 9G) and enhanced by reduced CCNB1 ($k_{s1}$) and increased CDH1P ($k'_{r11}$) (Fig 9H). By applying this integrated model of mitosis, our simulations have generated the hypothesis that HCMV-mediated inactivation of APC/C is necessary for producing a unique mitotic collapse defined by elevated levels of MPF kinase activity and LMNAP which is necessary for virion nuclear egress.

## Discussion

We have developed a novel integrated computational model of human mitotic cell cycle for the purpose of understanding how the human mitotic protein-protein interaction network (Fig 1A) is altered during disease. Unique features of this model include: (1) Integration of multiple pathways and regulations defining entry into, progression through, and exit from mitosis; (2) the role of PLK1 kinase in the regulation of mitosis; (3) inclusion of synthesis and process-dependent degradation of mitotic proteins; and (4) inclusion of the APC/C, CDC20 and CDH1 interactions. We have defined model parameters using a combination of previously published parameter values (directly adopted or fine-tuned) and that estimated based on analysis of quantitative and qualitative published data showing changes in key mitotic factors relative to each other over time (S4 Appendix). We have tested the model to reproduce the critical features of human mitosis as determined experimentally by numerous laboratories. Defining all relationships within a complex network is beyond the ability of any one laboratory. Unlike existing models, we have used a hybrid framework combining Michaelis-Menten and mass action kinetics for the mitotic interacting reactions (see S2 Appendix and S3 Appendix). The simulations approximate behaviors of protein interactions, direct and indirect, observed in synchronized human cell cultures and following perturbations that disrupt individual

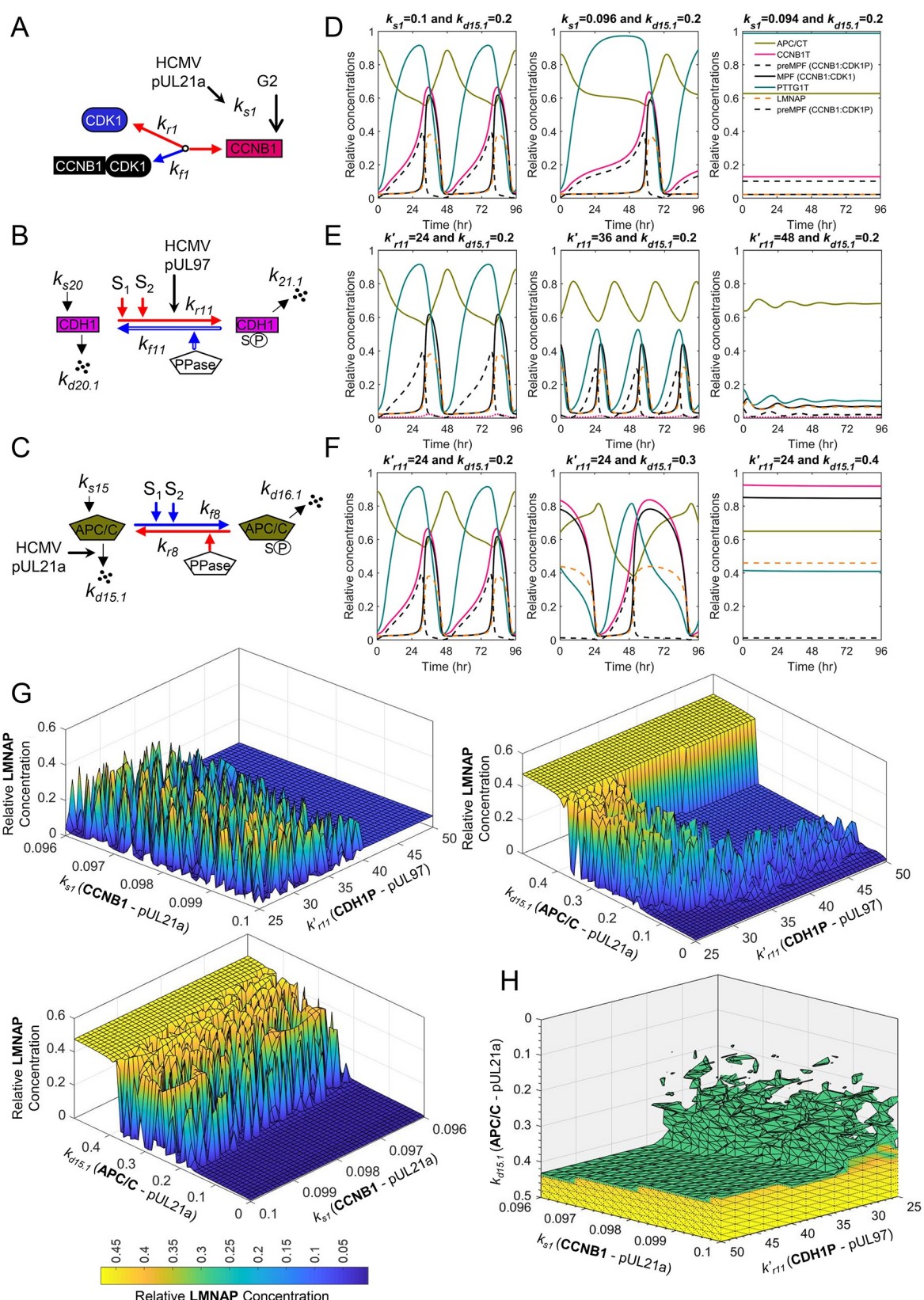

**Fig 9. A mitotic collapse resulting in sustained MPF activity during human cytomegalovirus infection requires APC/C subunit degradation.** Diagrams depicting the mechanisms of **(A)** HCMV pUL21a suppression of CCNB1 levels, **(B)** HCMV kinase pUL97 phosphorylation of CDH1, and **(C)** pUL21-mediated disruption of the APC/C complex during infection. **(D)** Simulated decrease in CCNB1 ($k_{s1}$) results in mitotic collapse defined by elevated levels of APC/CT and PTTG1T and loss of MPF (CCNB1:CDK1) and LMNAP. **(E)** Simulated increase in CDH1P ($k'_{r11}$) induces a collapse with elevated APC/CT yet loss of MPF, PTTG1, CCNB1 and LMNAP. **(F)** Increasing degradation of APC/C ($k_{d15.1}$) results in sustained MPF, LMNAP, and PTTG1T levels. **(G)** Simulating ranges of pUL21a and/or pUL97 activities in altering LMNAP levels at a single time point of 96 hr. Combined simulated increase in APC/C degradation ($k_{d15.1}$) with reduced CCNB1 synthesis ($k_{s1}$) result in low sustained levels of MPF and LMNAP which have been experimentally demonstrated to be required for virus replication and are the target of an antiviral agent. **(H)** Simulating ranges of pUL21a and pUL97 activities against CCNB1, APC/C and CDH1 and their impact on the levels of LMNAP as indicated by colored scale. Relative concentrations of LMNAP >0.3 are shown.

relationships. The result is a unique model of the mitotic biopathway defining relative changes over time between 12 different mitotic proteins and associated protein complexes using 15 major biochemical reactions and 26 ODEs. Unlike existing models, we have added rates for syntheses and degradations for different mitotic proteins which ultimately drive the mitotic oscillations. The simulation begins at mitotic entry upon increasing PLK1 kinase activity and CCNB1T levels resulting in formation of active MPF kinase (CCNB1:CDK1) (Fig 1B) and ends at mitotic exit following activation of the APC/CT:CDH1 ubiquitin ligase complex (Fig 1C).

We have constructed this base model of human mitosis in a modular format allowing for its future refinements and expansions. We have accounted for a subset of experimentally-defined mechanistic relationships. For example, we have included the activities of MPF (CCNB1: CDK1) and PLK1 kinases in converting CDC25C to CDC25CP. This is defined by the rate constant $k_{f3}$ involving $k'_{f3}$ (MPF) and $k''_{f3}$ (PLK1) (Fig 1A; Reaction 3, S2 Appendix; ODEs 7 and 8, S3 Appendix). However, additional mechanisms of regulation exist for this reaction. Our model does not distinguish between the contributions of different sites of phosphorylation in CDC25CP and their impacts on the mitotic cell cycle. PLK1 kinase phosphorylates cytosolic CDC25C on S198 resulting its translocation to the nucleus while MPF kinase modifies multiple residues (e.g. T138) of CDC25C resulting in increased CDC25CP activity. Likewise, PLK1 kinase modifies cytosolic MPF which stimulates its nuclear entry. These contributions can be added to the model such that total CDC25CP and total MPF kinase include both the cytosolic and nuclear pools. In addition, associated reactions and ODEs would then be modified to account for the compartmentalization. However, this distinction is transient in time due to increased LMNAP ($k'_{f15}$) and nuclear envelope breakdown (NEBD). Future work will be completed to account for these and other related factors and mechanisms. We also do not distinguish between the contributions of variants of CDC25C and other CDC25 family members to mitosis. Their contributions can be defined as specific signals (i.e. S1, S2, S3, etc.) within the governing ODEs, and considered in a similar fashion as done for the distinct contributions by MPF (S1) and PLK1 (S2) kinases. Many layers of regulation exist, and each additional mechanism will result in new reactions and ODEs integrated into the current model. The subsequent expansion will be aided by the increasing availability of datasets obtained from live-cell imaging using protein-specific sensors.

We have also included in the model the dynamic relationships between APC/C and coactivator subunits CDC20 and CDH1 (Fig 1A and 1C). The APC/C complex is autoinhibited until the subunits APC1 and APC3 become hyperphosphorylated. For this work, we do not distinguish between the contributions of individual APC/C subunits. Increased APC/CP is mediated by MPF (CCNB1:CDK1) and PLK1 kinases resulting in conformational changes and binding of unphosphorylated CDC20. Active APC/CP:CDC20 ubiquitin ligase promotes degradations of several substrates including CCNB1 and PTTG1 (Securin) (Fig 1A). In contrast, CDH1

binding is independent APC/C phosphorylation [43, 62] and we have separated APC/CT: CDH1 into APC/CP:CDH1 and APC/C:CDH1 (Fig 1A). Our simulation suggests that unphosphorylated APC/C is also involved in mitotic exit with CDH1 dissociating faster from APC/C:CDH1 than APC/CP:CDH1, indicating that APC/CP:CDH1 is more stable than APC/C: CDH1. The active APC/CT:CDH1 stimulates degradation of CDC20 and PLK1. Phosphatases (PPases) also play a vital role in mitotic progression. For oscillations in mitotic proteins to occur, we observed that PPase (i.e. PP2A)-regulated rate $k'_{f9}$ must be substantially greater than $k'_{r8}$ or $k'_{f11}$ (Fig 8B–8D). This observation confirms recently published experimental data by Hein *et al.* [13], demonstrating that temporal regulation (oscillation) is mediated by the preference of PP2A for dephosphorylating phosphothreonine residues over phosphoserine residues. Overall, our model incorporates unique features underlying the complex regulation of APC/C ubiquitin ligase that are required for the temporal progression through mitosis.

The primary purpose of developing this new integrated computational model is to generate new hypotheses and predict new experiments designed to study human disease. Disruptions to the mitotic network occur in diverse disease states including cancers and viral infections. In the simplest of terms, cancer is a disease of cell cycle dysfunction defined by loss of checkpoints and development of aneuploidy. Our computational model can simulate the DNA Damage Response (DDR) as well as the Spindle Assembly Checkpoint (SAC). Activation of CHK1 kinase upon DNA damage inhibits CDC25C while stabilization p53 inducing an arrest at G2/ M [63, 64]. Although the model does not explicitly account for CHK1 and p53 activities, we can simulate the response by making $k'_{r2} > k'_{f2}$ (inhibition of CDC25C) (Fig 1A) and $k_{s5} > 0.1$ (synthesis of CDKN1A) (Fig 6A). This same reaction is governed by the recently demonstrated G2/M metabolomic checkpoint involving AMPK and mTORC1, regulating CDC25C (increase $k'_{r3}$), CCNB1 (decrease $k_{s1}$) and PLK1 (decrease $k_{s12}$) (Fig 1A) [65, 66]. SAC involves sequestration of CDC20 to the mitotic checkpoint complex and away from APC/C. To simulate the checkpoint, we can set $k_{r10} > k_{f10}$ impacting the formation APC/CP:CDC20 (Fig 1A). Each of these processes are dysregulated in diverse cancers and an example can be seen in the plasma cell disorder, multiple myeloma (MM) [67]. Disrupting CDKs, PLK1, APC/C, and/or proteasome (i.e. bortezomib) activities in MM have all been identified as potential therapeutic targets. Our integrated computational model of the mitotic cell cycle will allow for *in silico* studies that investigate combinatorial approaches limiting MM proliferation and possibly inducing a mitotic collapse and/or catastrophe prior to extensive *in vitro* experimentation.

Numerous viruses also subvert the mitotic biopathway to support infection. A subset of viruses disrupt APC/C ubiquitin ligase complexes including HCMV, adenovirus, papillomavirus, and hepatitis B virus [68] with several of these associated with cellular transformation. For HCMV, the reason for APC/C inactivation and how it contributes to replication remain unknown. HCMV proteins manipulate the cell cycle immediately upon infection culminating in a mitotic collapse between 72 and 96 hrs post-infection [69]. The mitotic collapse is uniquely characterized by sustained CCNB1T, CDK1T and MPF kinase (CCNB1:CDK1) as well as elevated levels of CDC25CT and PTTG1T [69]. These factors and their activities have been included within our computational model (Fig 1A). Studies by Hamirally and Kamil *et al.* [52] have demonstrated that phosphorylation of LMNA contributes to nuclear egress of HCMV particles. The antiviral compound, maribavir inhibits this process and its antiviral activity is further supported by increased steady-state levels of CDKN1A (p21$^{CIP1}$) (Fig 6C) [51, 54, 70]. It is important to note that the kinetics of mitotic entry are regulated by the HCMV protein pUL21a which suppresses Cyclin A levels (Fig 9A) [55]. By simulating mechanisms of two viral proteins, pUL97 and pUL21a [56, 57, 59], we suggest that disrupting APC/C is necessary to induce a mitotic collapse supporting MPF kinase (CCNB1:CDK1) levels and nuclear lamina disruption (Fig 9H) [69]. There are numerous experimental perturbations that

result in a mitotic collapse exhibiting loss of MPF kinase activity (Figs 3F, 4D and 5E). These data and associated simulations suggest plausible targets for antiviral drug development that will be investigated in future studies.

We have constructed a novel comprehensive computational model that simulates the dynamics of mitotic biopathway. The reactions, ODEs and parameters have been validated using published experimental datasets and the model confirms the mechanisms regulating temporal progression through mitosis. Due to its modular format, which includes syntheses and degradations of mitotic proteins, we will be able to refine and/or extend the model in future work to include additional mechanisms of regulation and additional phases of the cell cycle. With the rapid discovery of cellular protein-protein networks and regulatory mechanisms, we anticipate that this model and the approach used in the development of this model will be highly valuable in helping us to understand network dynamics and identify potential points of therapeutic interventions.

## Materials and methods

### Experimental mitotic cell cycle datasets

Approximated changes in key mitotic regulators were obtained from several recently published datasets [7–9, 13, 38, 40, 49]. The information represents relative changes in specific factors (key mitotic regulatory proteins activities) over time and following a perturbation completed by different laboratories. For extracting data from immunoblot analyses, signal intensities were obtained from published PDF documents using Image Lab software (Bio-Rad Inc). Signals from gray scale images were inverted, and lanes and bands were identified manually. Signal intensities were obtained from the resulting signal volume as determined by the software. We refer to this information as approximated signal intensities and comparisons are limited to within datasets for individual proteins over time obtained from the same immunoblot. Data presented in line graphs were digitized using the freely available amsterCHEM ScanIt software (https://www.amsterchem.com/scanit.html). ScanIt is a program for extracting data from scientific graphs, particularly from line and scatter plots. When available, published quantitative values have been directly presented here as bar graphs. The biological significance of the extracted data and their utility for the model development are briefly described below.

Studies by Ovejero et al. [8] defined the contribution of CDC14A in the regulation of MPF and WEE1. To highlight key findings, approximated signal intensities were obtained from the published data, which used U2OS cells stably expressing HA-tagged WEE1. Ovejero et al. [8] synchronized cells at multiple stages and evaluated changes in specific mitotic factors by immunoblot analysis using lysates and following immunoprecipitation of WEE1-HA. Data on WEE1-HA stability were obtained using cycloheximide-treated cultures. The effects of disruption of CDC14A on specific mitotic factors were studied either by siRNA transfection of U2OS cells or by isolation of WEE1-HA from HEK293T cells followed by incubation with purified CDC14A. Signal intensities were determined as outlined above. In related studies, Potapova et al. [9] reported data investigating the importance of WEE1 and CDC25 in regulating mitosis. Specifically, Potapova et al. [9] determined changes in specific mitotic factors by immunoblot analysis following release of synchronized HeLa cells in the presence of CDC25 inhibitor NSC663285 and/or WEE1 inhibitor PD0166285. Signal intensities from immunoblots and line graphs from studies by Potapova et al. were acquired as outlined above.

The relative changes in PLK1 and WEE1 over time were determined in recent studies by Watanabe et al. [38] using immunoblot analysis of HeLa cell lysates isolated from synchronized cultures. Signal intensities from immunoblots were acquired as discussed above. Investigations on the roles of PLK1 and CDC25C in regulating mitosis were completed by Gheghiani

*et al.* [7]. Changes over time were analyzed using immunoblots following release of synchronized HeLa cells in the absence or presence of WEE1 inhibitor MK-1775. Gheghiani *et al.* also determined the impact of PLK1 inhibitor BI2536 on mitotic entry of HEK293 cells expressing wildtype or a phosphomimic version of CDC25C using live cell imaging. A subset of their data was obtained using ScanIt software and presented as bar graph as described above.

Studies by Harper *et al.* [49] provided the kinetic data regarding the decreased kinase activities of purified CCNB:CDK1 (MPF), CCNA:CDK2 and CCNE:CDK2 as functions of increasing concentrations of the kinase inhibitor CDKN1A (p21$^{CIP1}$). ScanIt software was used to digitize these data enabling us to build the kinetic models of CDKN1A binding to these kinases and estimate the relevant kinetic parameters (binding constants and Hill coefficients) characterizing these data and the inhibitory mechanisms of CDKN1A binding to these kinases.

The relative changes in APC/C and other related mitotic factors over time have been characterized by Zhao *et al.* [40]. In these studies, HeLa cells were synchronized in multiple phases and released, and changes in mitotic factors over time were determined by immunoblot analysis. Signal intensities were determined as outlined above. Studies by Hein *et al.* [13] demonstrated the distinct kinetics of dephosphorylation between CDC20 and CDH1 during transition through mitosis. In their studies, HeLa cells were depleted of endogenous CDC20 and transfected with either wildtype or 3SP mutant CDC20, and changes in fluorescent of CCNB1 were determined by live cell imaging. A subset of their data was obtained using ScanIt software. Hein *et al.* also determined difference in $^{32}$P labeling using purified MPF or PP2A against peptides of CDC20 wildtype or the 3SP mutant, as well as peptides of CDH1 wildtype or a 3TP mutant. These data were digitized using ScanIt software and presented as bar graphs.

## Model development, parameterization, and validation

Based on the aforementioned existing biological knowledge on the mitotic cell cycle regulations and the recent experimental data on the mitotic cell cycle (dys)regulations under diverse cellular perturbations, we established the human mitotic biopathway (biochemical reaction network), incorporating key mitotic proteins and their interactions, as schematized in Fig 1. Specifically, the biopathway incorporates the interactions of 12 unphosphorylated mitotic proteins (CDK1, CCNB1, WEE1, CDC25C, PLK1, PPase, CDKN1A, APC/C, CDC20, CDH1, PTTG1, and LMNA), 9 phosphorylated mitotic proteins (WEE1P, CDC25CP, PLK1P, PPaseP, APC/CP, CDC20P, CDH1P, PTTG1P, and LMNAP), and 6 mitotic protein complexes (preMPF, MPF, CDKN1A:MPF, APC/CP:CDC20, APC/C:CDH1, and APC/CP:CDH1). For nomenclature, we adopted the gene and protein names involved in the human mitotic cell cycle system based on information from the UniProt (Universal Protein) Resource, as detailed in S1 Appendix. Gene and protein names suffixed with P denote the phosphorylated proteins and that suffixed with T denotes the total protein.

The mitotic biopathway of Fig 1 incorporates 15 major biochemical reactions and other related reactions among 27 biochemical species (phosphorylated and unphosphorylated mitotic proteins and associated protein complexes), as detailed in S2 Appendix. The corresponding reaction rates are denoted by $k_{fn}$ and $k_{rn}$, where *f* represents forward direction (mitosis promoting), *r* represents reverse direction (mitosis arresting), and *n* represents reaction number (1, . . ., 15). These rates characterize either the elementary association and dissociation reactions or the phosphorylation and dephosphorylation reactions in the mitotic cell cycle system. We note here that $K_n = k_{rn}/k_{fn}$ is a constant when regulatory signals are not present in a reaction; $K_n$ determines the thermodynamic equilibrium of a reversible reaction and is equal to the mass action ratio, which is defined as the ratio of the product of the reactant concentrations and the product of the product concentrations at equilibrium.

In the mitotic biopathway of Fig 1, the synthesis rate of a protein is denoted by $k_{sn}$ (typically an unphosphorylated protein from $n = 1, \ldots, 27$) and degradation rate of a protein through a process is denoted by $k_{dn.m}$ ($n = 1, \ldots, 27$; $m = 1, 2$, and 3 denoting self-degradation, degradation mediated by APC/CP:CDC20, or degradation mediated by APC/CT:CDH1, respectively). The model considers 10 syntheses, 20 self-degradations, and 15 APC/C-dependent (APC/CP: CDC20, APC/CT:CDH1) degradations, as established experimentally, where APC/CT:CDH1 is the sum of APC/C:CDH1 and APC/CP:CDH1. We note here that the mitotic biopathway (Fig 1) does not distinguish between reactions occurring within the cytosolic and nucleus compartments.

The mitotic biopathway of Fig 1 was converted into a mathematical structure (i.e. dynamic integrated computational model) for the human mitotic cell cycle consisting of 26 ordinary differential equations (ODEs) describing the dynamics of the mitotic proteins (unphosphorylated and phosphorylated) and associated protein complexes during mitosis, as detailed in S3 Appendix. The ODEs are constructed based on the principle of mass conservation for the mitotic proteins/protein complexes using a hybrid Michaelis-Menten (MM) and mass action kinetic formulation for the mitotic interacting reactions. In the ODEs, [X] denotes the relative concentration of a protein X (normalized with respect to the total CDK1 concentration), $k_{sn}$ denotes the synthesis rate constants, $k_{dn.m}$ denotes the degradation rate constants, $k_{fn}$ denotes the forward reaction rate constants (cycle acceleration), and $k_{rn}$ denotes the reverse reaction rate constants (cycle deceleration). As mentioned above, the degradations occur via multiple processes (i.e. self and/or enzymatically mediated through APC/CP:CDC20 and/or APC/CT: CDH1), as established experimentally.

The mathematical formulation constitutes a total of 105 kinetic parameters in the model. All these parameters are difficult to estimate accurately from the semi-quantitative low-resolution experimental data available in the literature on the human/mammalian cell cycle regulations. So, we used an ad hoc approach of parameter estimation commonly used in the field of computational modeling of cell cycle [15–34]. Specifically, values of many of these parameters are directly adopted or fine-tuned from relevant published models in the literature from other laboratories [16, 19, 20, 23–26, 44, 45] (specifically [20] related to APC/C and CDC20 interactions), and the remaining parameter values are estimated based on scientific reasoning and thermodynamic and kinetic constraints (see S4 Appendix) to reproduce the cardinal features (some quantitatively and some qualitatively) of the human mitotic cell cycle based on recently published data (see Results). For given parameter values, to obtain the numerical solutions for the ODEs, we set the initial concentrations for the dephosphorylated proteins (i.e. CDK1, CDC25C, WEE1, PLK1, PPase, APC/C, CDC20, CDH1, PTTG1, LMNA) to 1 and the corresponding phosphorylated proteins and all other proteins and protein complexes to 0, including CCNB1 and CDKN1A (p21$^{CIP1}$). We note here that the 26 ODEs describe the dynamic alterations in the protein activities (i.e. relative concentrations normalized with respect to the total CDK1 concentration), which are non-dimensional variables (unitless). Integrating the 26 ODEs provides the dynamic simulations that characterize the mitotic entry, anaphase transition, and mitotic exit. Additional phases of the cell cycle and associated protein dynamics and cellular compartmentalization are not accounted for in this model of the human mitotic cell cycle. Furthermore, mechanisms of regulations (proxy signals regulating the mitotic reactions) occurring in the G1, G2 and G2/M phases are not accounted for in this model and will be considered in future studies.

## Supporting information

**S1 Appendix. The gene and protein names involved in the human mitotic cell cycle system.**
(DOCX)

**S2 Appendix. The major biochemical reactions and other related reactions among the mitotic proteins and the associated protein complexes and their descriptions based on the mitotic biopathway of Fig 1.**
(DOCX)

**S3 Appendix. The ODEs (ordinary differential equations) that describe the dynamics of the mitotic proteins and the associated protein complexes based on the principle of mass conservation.**
(DOCX)

**S4 Appendix. Table of model parameters (descriptions, values, and references).**
(DOCX)

**S5 Appendix. Model simulations demonstrating oscillations of human mitotic proteins and protein complexes during the progression of cell cycle for two different periods of oscillation (48 hrs and 24 hrs), controlled by a time scale parameter α.**
(DOCX)

## Author Contributions

**Conceptualization:** Scott S. Terhune, Ranjan K. Dash.

**Data curation:** Scott S. Terhune.

**Formal analysis:** Scott S. Terhune, Yongwoon Jung, Ranjan K. Dash.

**Funding acquisition:** Scott S. Terhune, Ranjan K. Dash.

**Investigation:** Scott S. Terhune, Yongwoon Jung, Katie M. Cataldo, Ranjan K. Dash.

**Methodology:** Scott S. Terhune, Yongwoon Jung, Ranjan K. Dash.

**Supervision:** Scott S. Terhune, Ranjan K. Dash.

**Validation:** Scott S. Terhune, Ranjan K. Dash.

**Visualization:** Scott S. Terhune, Yongwoon Jung.

**Writing – original draft:** Scott S. Terhune, Yongwoon Jung, Ranjan K. Dash.

**Writing – review & editing:** Scott S. Terhune, Ranjan K. Dash.

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
