## [Decision Letter · Decision Letter 0]

19 Aug 2019

Dear Dr Dash,

Thank you very much for submitting your manuscript 'Network Mechanisms and Dysfunction within an Integrated Computational Model of the Human Mitotic Cell Cycle' for review by PLOS Computational Biology. Your manuscript has been fully evaluated by the PLOS Computational Biology editorial team and in this case also by independent peer reviewers. The reviewers appreciated the attention to an important problem, but raised some substantial concerns about the manuscript as it currently stands. While your manuscript cannot be accepted in its present form, we are willing to consider a revised version in which the issues raised by the reviewers have been adequately addressed. We cannot, of course, promise publication at that time.

Sincerely,

John J. Tyson

Guest Editor

PLOS Computational Biology

Jason Haugh

Deputy Editor

PLOS Computational Biology

[LINK]

Your paper has been read and critiqued by three referees with expertise in cell cycle modeling and viral dynamics. All three referees recognize and admire the large amount of work you have put into developing this comprehensive model of the molecular network controlling progression through the human cell cycle, and they would like to see it published. But they have made many cogent suggestions for improvement of the model and its presentation. In a revised version (which is encouraged by this guest editor), the authors should describe more explicitly what makes their model unique and especially appropriate for describing human cell cycle regulation. They also must provide more evidence for constraining the values of the kinetic parameters in the model by comparison to quantitative data on mRNA and protein levels during progression through the human cell cycle. Nowadays, to make quantitative predictions of the outcome of novel experiments using a mathematical model, it is customary to predict results using an ensemble of "acceptable" sets of rate parameter values (rather than a single "best" set, or, worse still, a set of parameter values many of which are arbitrary).

One referee suggests that the authors provide a simpler model, with fewer equations and parameters, that specifically addresses the effects of viral infection on cell cycle progression. I will not insist on this, but the authors should address the question (in the discussion?) of what specific aspects of the general model are relevant to the issue of response to viral infection. The authors should also consider other applications of the model to justify the utility of their comprehensive model.

As customary, along with the revised submission, the authors should provide a detailed discussion of how they have addressed (or why they have not addressed) every substantive comment of the referees.

Reviewer's Responses to Questions

**Comments to the Authors:**

Reviewer #1: This study proposes an integrated computational model to better understand the molecular network driving the human mitotic cell cycle and its deregulation during pathogenesis. The model accounts for the effect of human herpesvirus cytomegalovirus. In particular, it predicts that the virus-mediated disruption of APC/C is necessary to sustain a unique mitotic collapse defined by stable MPF levels.

The article is interesting and well written. However, I have some concerns that need to be addressed before I could recommend the manuscript for publication:

- Your model is able to simulate Potapova experiments showing the impact of CDC25C and WEE1 regulation on MPF (Fig. 5). A previous mathematical model already reproduced Potapova experiment (Tuck C. et al. Robust mitotic entry is ensured by a latching switch. Biol. Open 2013). What is the adding value of your model as compared to this previous model? I believe it should be discussed.

- You have developed an exhaustive model of the human mitotic cell cycle which incorporates 105 parameters. I am convinced that this approach is useful to better highlight the qualitative dynamics of a regulatory structure, but I am not convinced that you can make strong quantitative predictions. In particular, you predict that the virus-mediated disruption of APC/C is necessary to sustain a unique mitotic collapse. This prediction may be dependent of your specific set of parameter values. The model may exhibit complete different dynamics if you are located in a different region of your parameters space.

- You mention that the computational model is for the human mitotic cell cycle because you consider only regulations present in humans. Have you calibrated our model quantitatively on human data? Nowadays, mRNA and protein expression levels of numerous gene can be available for numerous human cancers (i.e. The Cancer Genome Atlas and The Cancer Proteome Atlas: https://tcpaportal.org/tcpa/). This is the same for human cancer cell lines. In addition, half-life durations of mRNAs and proteins can be available in the literature. All the protein data could help you to calibrate your model in a more precise manner, which could reinforce the model’s predictions.

- The wiring diagram in Figure 1 is exhaustive but very complex. In addition to this scheme, I would like to see a more simplified wiring diagram where the regulatory structure of the network, i.e. positive feedback loops, is better highlighted.

- I do not think that the time series in Figure 2 is very informative. I propose to remove it or to put it in Supplementary Information.

Reviewer #2: Terhune et al. model the phosphorylation status and concentration of key oscillating cell cycle proteins during mitosis. They develop a comprehensive model and clearly a substantial amount of work went into this manuscript. The study relies on mass action kinetics to model kinetic interactions and uses Michaelis-Menten behavior to model enzyme kinetics. The authors seek to coalesce the myriad cell cycle protein interactions in the literature to build a model capable of not only mirroring key features observed in mitosis, but also providing insight into novel circuits that may appear under viral stress. We applaud the approach of trying to build a comprehensive model, but there is also a downside with respect to tractability of the model and whether such a model can generate unique and falsifiable predictions. The latter point needs to be addressed more clearly. We would prefer a reductionist model, if the same physiological behaviors can be achieved via such a simpler model.

While the study is comprehensive and appears careful, to enhance the study’s contribution to the field we have the following comments.

Major comments:

1. Far too many parameters (85 of the 105) are arbitrarily assigned in order to recapitulate cardinal features of mitosis. This brings the validity of the model into question. Further justification for these parameter values is needed. Non-dimensionalization could reduce the number of parameters but this will likely only go so far. Course graining and sensitivity analysis (i.e., lumping together reactions or parameters) would provide an idea of how critical specific reactions are to the behavior. We think a simpler more tractable model is needed.

2. A regression fit to known datasets for key protein concentrations kinetics would add more validity to the model and would allow the authors to reduce the model to key state variables that can be fit. Kinetic Data and model fits over time, shown on the same figure, is common way to present this and should be shown.

3. Experimentally testable predictions of the model should be clearly laid out in a subsection (perhaps in the discussion). This includes the prediction that Kf9 is >> Kr8 and Kf11. This preference of PP2A for specific reactions could be tested easily in vitro by performing kinetic analysis. Although there are no clear predictions made from the model under the various viral stresses simulated (if there are, it would be nice to have them listed together at the end of the discussion section), one could verify the parameter space that induces mitotic collapse in mitotically trapped cells by treating them with increasing doses of the perturbed protein(s) (in Figure 10 simulations) in cell culture. One could also propose decreasing key proteins in a dose dependent manner.

4. Typically, the most significant contribution of a model to a field is either (a) explanation of a phenomenon that was unclear or (b) prediction of new experiments that generate non-intuitive outcomes or ideally (c) both. The authors should make clear (in the abstract) in which of these ways the model contributes and then provide background for what was unclear in the field if (a) or (c).

Specific comments:

1. Figure 1 is very confusing. Key reactions aren’t displayed; specifically, the MPF phosphorylation of the APC/C complex is left out. The labelling scheme to separate reactions by the order they occur is not visually pleasing or intuitive. A left to right approach with respect to the order of events would be better. One could also separate reactions that occur during a particular stage into boxes and have lines cross the boxes for any transitional interactions.

2. The MPF complex is not defined clearly in the text or Figure 1. It is regularly interchanged for CCNB1:CDK1. Consistency should apply, and only one name should be used for the complex after it has been defined as consisting of unphosphorylated CCNB1 and CDK1.

3. Would like to see if the same behaviors observed in this model can be seen with a simpler model without so many unknown parameters (i.e., reduce the number of “#” in the Table in the supp info).

Reviewer #3: The authors developed a mathematical model for human mitotic cell cycle and use the model to investigate dysfunction in pathogenesis.

General comments:

Throughout the manuscript, the authors stated a couple of times that they developed a novel integrated computational model for human mitotic cycle. However, by reading through whole manuscript, I had difficult to find the novelty and the authors did not state clearly what novel aspects are in their study. For example, what new interactions and parameters have included in this model comparing to the many previous models of cell cycle, in particular the mammalian cell cycle models? Are there new bistable switches in the new model in addition to the known ones? What the are the new insights gained via current modeling into the experimental observations? I understand that the authors are developing a human model, which could be novel since no human cell cycle model has been developed. But it needs to state clearly what special aspects of model justify it as a human model, not another mammalian cell cycle model, or the novel mechanistic insights that the model provides for the human mitotic cell cycle. I appreciate the efforts that authors had put to develop this complex model, it should be made clear to the readers what is new in the paper.

Specific comments:

1. The first sentence in the Abstract is somewhat awkward by saying that “Our ability to understand … is becoming more difficult due to …” It is becoming more difficult is due to the discovery of more and more new protein interactions and mechanisms. Our ability and the mechanisms are not becoming difficult.

2. The abbreviations in the Abstract are not defined.

3. The authors should state clearly in the Introduction what specific problems they want to solve or the specific rationale for them to develop the human model.

4. The different arrow types in Fig.1 is very confusing and difficult to digest. For example, protein degradation (e.g., Cyclin B) may occur all the time during the cell cycle, more in one phase while less in another phase. But using solid, open, and dashed arrows to indicate degradation in different phases makes it difficult to read. Another example: APC/C activity peaks at the anaphase, which is indicated by the open arrows, but MPF also peaks at the anaphase, but it is not marked by open arrows.

5. Too many lines plotted in a single panel Fig.2, you may plot them in different panels by subgroups. Looks like that Fig.3 is not new but regrouping of Fig.2.

6. Based on Fig.2, the cycle time is 48 hours. Should be human mitotic cell cycle period around 24 hours?

7. I find that it is very difficult to compare the experimental data with the simulation results in Figs. 4, 5, 6 and 8. The author should use the same bar graphs as for the experimental data for comparison.

8. Except for some parameters were taken from previous studies, mainly modeling studies, most of the parameters were set arbitrarily and unitless. Can you determine some of the parameters by comparing the simulation results with the experimental data quantitatively? I think that this is one important aspect to justify your model as a human model since the parameters are determined based on human data.

**Have all data underlying the figures and results presented in the manuscript been provided?**

Reviewer #1: Yes

Reviewer #2: Yes

Reviewer #3: Yes

PLOS authors have the option to publish the peer review history of their article (what does this mean?). If published, this will include your full peer review and any attached files.

Reviewer #1: No

Reviewer #2: No

Reviewer #3: No

---

## [Decision Letter · Decision Letter 1]

25 Jan 2020

Dear Dr. Dash,

Thank you very much for submitting your manuscript "Network Mechanisms and Dysfunction within an Integrated Computational Model of the Human Mitotic Cell Cycle" for consideration at PLOS Computational Biology. As with all papers reviewed by the journal, your manuscript was reviewed by members of the editorial board and by several independent reviewers. The reviewers appreciated the attention to an important topic. Based on the reviews, we are likely to accept this manuscript for publication, providing that you modify the manuscript according to the review recommendations.

Two of the three original reviewers find the revised version acceptable for publication. Reviewer #2 did not submit an opinion on the revised paper. The guest editor is willing to accept the revised paper for publication subject to some minor revisions:

1. The title is misleading because the authors have not modeled the complete cell cycle in human cells (no description of progression through G1 and entry into S phase). Rather, they are only describing progression from G2 into mitosis, and then out of mitosis into G1. Their model equations show limit cycle oscillations only because their G1-state morphs into a G2-state without doing DNA replication. This is OK, because the authors only want to describe in detail the molecular changes occurring as a human cell transits into and out of mitosis. But the text should make this clear. And the title should be changed to "... an Integrated Computational Model of Progression through Mitosis in the Human Cell Cycle".

2. In Fig. 1 the notation 'S1' and 'S2' is a little confusing at first sight. The legend of the figure should say explicitly that S1 is active MPF (CCNB1:CDK1) and S2 is active (phosphorylated) PLK1.

3. Regarding 'cell cycle time of 48 h', I think the whole discussion on p 13 is misguided. Because the authors are not simulating the entire human cell cycle (G1-S-G2-M), the period of oscillation of their model equations is unrelated to observed cell division times in human cells. The only relevant parts of their simulations are how the cell enters into M from late G2, and exits from M into early G1. The parameter alpha in their ODEs (Appendix S3) can be adjusted to make this time interval fit the facts in mammalian cells. I don't know what the time period is from late G2 to early G1 in mammalian cells in culture, but I bet it is much shorter than the 50 h in Fig 2.

4. The authors seem to suggest that they can fit their model to any cell cycle duration simply by changing the time-scale factor, alpha. This is true mathematically, but it doesn't make any sense biologically. Cell cycle duration is not adjusted by scaling all rate constants up or down by a constant factor! Rather, cell cycle duration is determined by factors outside the cell-cycle signaling network: by things like nutrient availability, oxygenation, growth factors, cell-cell contacts, etc.

I would like the authors to make some small adjustments to the revised version to take into account these comments.

Sincerely,

John J. Tyson

Guest Editor

PLOS Computational Biology

Jason Haugh

Deputy Editor

PLOS Computational Biology

[LINK]

Two of the three original reviewers find the revised version acceptable for publication. Reviewer #2 did not submit an opinion on the revised paper. The guest editor is willing to accept the revised paper for publication subject to some minor revisions:

1. The title is misleading because the authors have not modeled the complete cell cycle in human cells (no description of progression through G1 and entry into S phase). Rather, they are only describing progression from G2 into mitosis, and then out of mitosis into G1. Their model equations show limit cycle oscillations only because their G1-state morphs into a G2-state without doing DNA replication. This is OK, because the authors only want to describe in detail the molecular changes occurring as a human cell transits into and out of mitosis. But the text should make this clear. And the title should be changed to "... an Integrated Computational Model of Progression through Mitosis in the Human Cell Cycle".

2. In Fig. 1 the notation 'S1' and 'S2' is a little confusing at first sight. The legend of the figure should say explicitly that S1 is active MPF (CCNB1:CDK1) and S2 is active (phosphorylated) PLK1.

3. Regarding 'cell cycle time of 48 h', I think the whole discussion on p 13 is misguided. Because the authors are not simulating the entire human cell cycle (G1-S-G2-M), the period of oscillation of their model equations is unrelated to observed cell division times in human cells. The only relevant parts of their simulations are how the cell enters into M from late G2, and exits from M into early G1. The parameter alpha in their ODEs (Appendix S3) can be adjusted to make this time interval fit the facts in mammalian cells. I don't know what the time period is from late G2 to early G1 in mammalian cells in culture, but I bet it is much shorter than the 50 h in Fig 2.

4. The authors seem to suggest that they can fit their model to any cell cycle duration simply by changing the time-scale factor, alpha. This is true mathematically, but it doesn't make any sense biologically. Cell cycle duration is not adjusted by scaling all rate constants up or down by a constant factor! Rather, cell cycle duration is determined by factors outside the cell-cycle signaling network: by things like nutrient availability, oxygenation, growth factors, cell-cell contacts, etc.

I would like the authors to make some small adjustments to the revised version to take into account these comments.

Reviewer's Responses to Questions

**Comments to the Authors:**

Reviewer #1: Dear authors,

I believe the manuscript is well improved. I accept the manuscript for publication.

Reviewer #3: no more comments

**Have all data underlying the figures and results presented in the manuscript been provided?**

Reviewer #1: Yes

Reviewer #3: Yes

PLOS authors have the option to publish the peer review history of their article (what does this mean?). If published, this will include your full peer review and any attached files.

Reviewer #1: No

Reviewer #3: No
---

## [Editor Report · Decision Letter 2]

12 Feb 2020

Dear Dr. Dash,

We are pleased to inform you that your manuscript 'Network Mechanisms and Dysfunction within an Integrated Computational Model of Progression through Mitosis in the Human Cell Cycle' has been provisionally accepted for publication in PLOS Computational Biology.

Before your manuscript can be formally accepted you will need to complete some formatting changes, which you will receive in a follow up email. A member of our team will be in touch within two working days with a set of requests.

Best regards,

John J. Tyson

Guest Editor

PLOS Computational Biology

Jason Haugh

Deputy Editor

PLOS Computational Biology

the authors have adequately responded to the remaining "minor revisions" requested by the reviewer. So the paper is now acceptable for publication.

---

## [Editor Report · Acceptance letter]

27 Mar 2020

PCOMPBIOL-D-19-00988R2 

Network Mechanisms and Dysfunction within an Integrated Computational Model of Progression through Mitosis in the Human Cell Cycle

Dear Dr Dash,

I am pleased to inform you that your manuscript has been formally accepted for publication in PLOS Computational Biology. Your manuscript is now with our production department and you will be notified of the publication date in due course.

With kind regards,

Laura Mallard
